# Unique genetic and risk-factor profiles in clusters of major depressive disorder-related multimorbidity trajectories

Andras Gezsi [1,13], Sandra Van der Auwera [2,3,13], Hannu Mäkinen[4],
Nora Eszlari[5,6], Gabor Hullam[1,5], Tamas Nagy[1,5,6], Sarah Bonk[2],
Rubèn González-Colom [7], Xenia Gonda [5,6,8], Linda Garvert [2],
Teemu Paajanen [4], Zsofia Gal [5,6], Kevin Kirchner[2], Andras Millinghoffer[9],
Carsten O. Schmidt [10], Bence Bolgar[1], Josep Roca[7], Isaac Cano [7],
Mikko Kuokkanen [4,11,12], Peter Antal [1,14] & Gabriella Juhasz [5,6,14] ✉

The heterogeneity and complexity of symptom presentation, comorbidities and genetic factors pose challenges to the identification of biological mechanisms underlying complex diseases. Current approaches used to identify biological subtypes of major depressive disorder (MDD) mainly focus on clinical characteristics that cannot be linked to specific biological models. Here, we examined multimorbidities to identify MDD subtypes with distinct genetic and non-genetic factors. We leveraged dynamic Bayesian network approaches to determine a minimal set of multimorbidities relevant to MDD and identified seven clusters of disease-burden trajectories throughout the lifespan among 1.2 million participants from cohorts in the UK, Finland, and Spain. The clusters had clear protective- and risk-factor profiles as well as age-specific clinical courses mainly driven by inflammatory processes, and a comprehensive map of heritability and genetic correlations among these clusters was revealed. Our results can guide the development of personalized treatments for MDD based on the unique genetic, clinical and non-genetic risk-factor profiles of patients.

Major depressive disorder (MDD), characterized by depressed mood and loss of interest, is one of the most common mental disorders and typically has a recurrent or chronic course that leads to suffering, disability, increased suicide risk, and increased all-cause mortality[1]. Additionally, approximately one-third of patients are resistant to current treatments[2], partly due to the lack of a comprehensive biological model, as MDD shows remarkable heterogeneity in both its clinical manifestation and underlying neurobiology[3].

Genome-wide association studies (GWASs) have suggested divergent pathways and mostly nonspecific cellular processes involved in MDD. Furthermore, genetic correlation studies have indicated shared genetic contributions to several somatic and mental disorders,

reflecting the pleiotropy of genetic variants and common biological processes involved[4–6]. As twin studies have estimated that only 40% of phenotype heritability can be attributed to additive genetic effects[6], this suggests the involvement of comorbid conditions and non-genetic risk factors.

The inclusion of additional phenotypic information when subtyping depressive disorders can indeed reduce genetic heterogeneity and inform the development of aetiological models, even if such subtyping was based on restricted sets of predefined criteria and associated symptoms[7]. Furthermore, a recent study demonstrated strong multimorbidity patterns among 439 common diseases on both phenotypic and genetic levels[8], revealing strong correlation patterns

A full list of affiliations appears at the end of the paper. . ✉e-mail: juhasz.gabriella@semmelweis.hu

between psychiatric and cardiovascular and respiratory disorders. The multimorbidity paradigm shift also highlights the importance of considering multimorbidity patterns when identifying depression subtypes and determining distinct biological pathways in their background[9,10].

Thus, in the *Temporal Disease Map-Based Stratification of Depression-Related Multimorbidities* (TRAJECTOME) project, our aim is to filter and include all relevant information on trajectories of MDD-related multimorbidities from large population cohorts, including individuals with and without MDD, to facilitate the identification of biologically and clinically informative depression subtypes as well as their distinct neurobiological and genetic backgrounds, and thereby suggest potential biomarkers for precision screening and treatment. Our main hypothesis was that the use of age-dependent strongly relevant MDD-related multimorbidities enriches the genetic basis of MDD, such that specific participant clusters are associated with distinct genetic profiles contributing to the pathology of major depressive disorder.

To date, studies attempting to identify subgroups within chronic somatic diseases have mainly used cross-sectional data and have not considered associations with psychiatric disorders[11–13]. Recently, this approach was expanded from phenotypic data to genetic data[8], explaining 46% of identified multimorbidities according to shared genetic components and identifying central hub diseases that are highly relevant to the majority of multimorbidity relationships. Furthermore, a Danish group identified five time-dependent psychiatric-multimorbidity clusters in schizophrenia patients associated with heterogeneity in aetiological factors[14].

Moving beyond these attempts, our approach is based on dynamic Bayesian networks to identify a minimal, nonredundant set of MDD-related multimorbidity trajectories that convey all relevant information about MDD in an individual's entire medical history. This filtered set includes all multimorbidities with nonmediated causal relationships to MDD and those with potential shared genetic and non-genetic factors. In a previous cross-sectional analysis, we demonstrated that such Bayesian filtering of pairwise comorbidity associations significantly boosted the shared molecular background[15]. In this project, we leveraged this enrichment effect and extended our previous approach into the temporal dimension. The method involved three steps. First, we utilized the statistical concept of the Markov boundary to identify the minimal set of multimorbidity trajectories relevant to a target variable (MDD). Second, we developed a unique, data-driven method of measuring patient similarity in this filtered set of MDD-related trajectories. Finally, we used privacy-preserving Bayesian federated clustering of individuals in a transcohort setting: we used five general-population cohorts ($N = 1,189,509$) for discovery and two additional cohorts ($N = 387,089$) to validate the clusters' associated genetic profiles and non-genetic risk factors.

This approach provides a temporal, systems-based perspective on the complex pattern of time- and comorbidity-dependent courses of MDD and their associated biological risk factors, yielding a distilled molecular understanding of the disease network and possibly indicating novel long-term treatment approaches. Moreover, our approach enables the dissection of shared genetic factors between MDD and related conditions and may pave the way for personalized treatment plans targeting not diseases but specific shared pathways in multimorbid conditions, particularly in the realm of psychiatric disorders.

## Results

The TRAJECTOME project involves data from 1,576,598 participants from seven European general-population cohorts (Table 1, Supplementary Data 1A, B). To identify distinct MDD-related multimorbidity-based clusters and assess their biological profiles, we used individual disease onset information from large cohorts divided into discovery and validation cohorts. The observed differences between cohorts included age range, birth year, and socioeconomic factors, which may have influenced the availability of medical care and disease diagnosis and affected the prevalence rates of lifetime MDD diagnosis (7–19%) in the cohorts (Supplementary Figs. S1 and S2).

### Dynamic Bayesian network analysis reveals seven MDD-related multimorbidity clusters

To identify MDD-related multimorbidity clusters from the discovery cohorts' temporal trajectories, we selected 86 predetermined *cross-cohort diseases* strongly related to MDD (Methods, Fig. 1, Supplementary Data 2 and 3). Based on these diseases, we computed weighted direct MDD-related multimorbidity scores for each participant in each cohort over different time intervals and used these scores as input in the cluster analysis. The seven identified clusters reflected different *temporal trajectories of the MDD-related multimorbidity burden* throughout the lifespan (Supplementary Fig. S3.1) and corresponded to specific clinical subtypes, which we characterized according to genetic profiles and non-genetic risk factors. The fundamental hypothesis of our method was that defining the clusters based only on diseases strongly relevant to MDD would focus the clusters' profiles on pleiotropic genetic and non-genetic factors of these diseases (Fig. 1). In contrast, the influence of factors affecting MDD only through other diseases was diminished (such indirectly related diseases are associated with but not strongly relevant to MDD[15]). The resulting clusters represent special multitraits and, as such, combine evidence from the multimorbidities strongly relevant to MDD and enrich their common influencing factors. Therefore, our framework represents a more powerful method of identifying shared mechanisms influencing MDD and each identified clinical subtype.

To determine the clinical characteristics of the identified clusters, we investigated their temporal disease patterns in terms of the mean and distribution of onset age of each cross-cohort disease (Fig. 2). We also evaluated the disease risk based on Cox regression for each cross-cohort disease and the MDD-free survival using Kaplan–Meier estimates (Fig. 3).

Regarding the mean onset age of the cross-cohort diseases in the UK Biobank (UKB) cohort (Fig. 2), Clusters 1–4 had a later onset age and a longer period of low disease burden. Cluster 6 was similar but had higher disease prevalence in older age. In Cluster 5, the mean onset ages of diseases, especially musculoskeletal, respiratory, and genitourinary diseases, were earlier, whereas in Cluster 7, the mean onset ages of allergic and respiratory inflammatory diseases, migraine, and dermatitis were earlier. Thus, the onset ages of comorbid diseases in Cluster 7 exhibited a bimodal distribution (Fig. 2B), with the first peak occurring before the age of 20 years and the second peak, which reflected age-related disorders, occurring later, comparable to the peak of Cluster 6 (complete trajectories of all cross-cohort diseases as well as comparisons among the cohorts are provided in Supplementary Figs. S3.2 and S4.1–S4.4). The distribution of the onset ages (Fig. 2B) among the clusters is also reflected in their age distributions. Clusters 5 and 7 mainly included younger individuals assigned at early ages of disease onset (Supplementary Fig. S5), whereas the remaining clusters mainly included older individuals who were only assigned with high certainty after the first diseases had emerged at an older age.

Concerning the prevalence of comorbid diseases, we found a clear distinction in the burden of MDD-related disorders among the clusters (Fig. 3A). Clusters 1–4, and especially Clusters 1–2, exhibited a low prevalence of almost all cross-cohort diseases. A substantial decrease in the prevalence rates of psychiatric and respiratory diseases was observed along with slightly increased prevalence rates of cerebrovascular and kidney diseases and hypertension (in Cluster 3) or of lipid metabolic disorders and hypothyroidism (in Cluster 4). Clusters 5–6 exhibited a higher MDD-related disease burden with increased prevalence rates of almost all cross-cohort diseases (Fig. 3A). In Cluster

**Table 1 | Characteristics of the individual cohorts included in the TRAJECTOME project**

| | UKB (N = 502,504) | THL (Finrisk, Health 2000/2011, FinHealth 2017) (N = 41,092) | CHSS (N = 645,913) | FinnGen (N = 385,640) | SHIP (N = 1449) |
|---|---|---|---|---|---|
| Analysis status | Discovery[a] | Discovery | Discovery | Validation | Validation |
| Available information types (clinical, genetic, behavioural factors) | (Yes, Yes, Yes) | (Yes, Yes, Yes) | (Yes, No, No) | (Yes, Yes, No) | (No, Yes, Yes) |
| Country | UK | Finland | Spain | Finland | Germany |
| Age, years | 61.5 (9.31), [37–83] | 65.1 (14.02), [21–99] | 47.5 (24.36), [0–111] | 59.4 (18.3), [0–99] | 60.4 (12.8), [36–93] |
| Sex, Male Female | 229,122 (45.6%) 273,382 (54.4%) | 19,186 (46.7%) 21,906 (53.3%) | 306,337 (47.4%) 339,576 (52.6%) | 169,103 (43.8%) 216,537 (56.2%) | 660 (45.5%) 789 (54.5%) |
| Education <10 years = 10 years > 10 years | 95,403 (19.0%) 132,084 (26.3%) 275,017 (54.7%) | NA | NA | NA | 307 (21.2%) 822 (56.7%) 317 (21.9%) NA: 3 (0.2%) |
| Year of birth | 1951 (8.1), [1934–1971] | NA | 1971 (24.4), [1894–2019] | NA | 1954 (12.9), [1921–1978] |
| Household income Low Medium High | 117,737 (23.4%) 358,492 (71.3%) 26,275 (5.2%) | 14,787 (36.0%) 13,593 (33.0%) 12,712 (31.0%) | 372,238 (57.6%) 249,670 (38.7%) 24,005 (3.7%) | NA | 183 (12.6%) 1078 (74.4%) 188 (13.0%) |
| Lifetime MDD (F32 and F33 combined) | 53,473 (10.64%) | 3014 (7.33%) | 47,162 (7.3%) | 46,153 (12.0%) | 382 (26.4%) |

For continuous variables, the mean (standard deviation) and range are given; for categorical variables, the counts and percentages are given.

*NA* not applicable.

[a]Primary analyses were all performed in the UKB cohort, as this was the largest cohort with all information types available. Included cohorts: the UK Biobank (UKB), Finnish Institute for Health and Welfare cohorts (THL (Finrisk, Health 2000 & 2011, FinHealth 2017)), Catalan Health Surveillance System (CHSS), FinnGen project (FinnGen), and Study of Health in Pomerania (SHIP). For more details, see the Methods.

5, the prevalence rates of schizophrenia and musculoskeletal diseases leading to pain disorders were increased, while in Cluster 6, the prevalence rates of severe reactions to stress, somatoform disorders, and respiratory tract infections were increased. Cluster 7 had a divergent disease profile, with slight decreases in the prevalence rates of most disorders, except for allergic and respiratory inflammatory diseases, migraine, and dermatitis, which each had a strongly increased prevalence (Fig. 3A).

Regarding the evaluation of MDD, the same disease-burden pattern in terms of onset age and prevalence was observed in all five cohorts (Fig. 3B, C), with Clusters 1–4 having a low MDD burden in contrast to Clusters 5–7 having a high MDD burden with greater variations among the cohorts. Focussing on the complete set of psychiatric diseases (Chapter V. of ICD-10: F00-F99) to assess their temporal disease patterns across the identified clusters, the analysis revealed significant differences among the clusters in terms of the onset and prevalence of various psychiatric disorders (Supplementary Fig. S6).

This pattern was also reflected in the correlations of cluster membership probabilities throughout all cohorts (Supplementary Fig. S7), with strong positive correlations of cluster membership among Clusters 1–4 and between Clusters 5 and 6 and mainly negative correlations of cluster membership with Cluster 7, reflecting the three divergent risk profiles. Distributions of the cluster membership probabilities in the UKB cohort (Supplementary Figs. S8.1–S8.7) and of the final cluster assignment in all cohorts (Supplementary Fig. S9) showed that most participants were assigned to one of the low-risk Clusters 1–4. Within the remaining risk-conferring clusters, the majority of individuals were assigned to the early-onset clusters (Clusters 5&7), and the fewest individuals were assigned to the late-onset Cluster 6. These findings suggest that the identified clusters had distinct clinical characteristics, which could have implications for personalized healthcare approaches, early intervention strategies[16], and targeted treatment plans for individuals within each cluster.

## GWAS analysis of MDD-related multimorbidity clusters in the UKB cohort identifies immune system-related genetic profiles

To explore the genetic contribution of the clusters, we conducted GWAS analyses in the UKB cohort (N = 249,167), where the posterior log-odds of the cluster memberships were used as the target variables. Analyses of all clusters revealed 6141 distinct genome-wide significant single-nucleotide polymorphisms (SNPs) spanning 42 risk loci on 20 different chromosomes (Table 2). Individual Manhattan and QQ plots (Fig. 4A, Supplementary Figs. S10.1–S10.7) and genomic risk loci (Supplementary Data 4–10), gene-based (Supplementary Data 11–18), and functional enrichment analyses results (Supplementary Data 19) are provided in the Supplement.

The overall pattern of risk-conferring and protective clusters was also apparent at the genetic level (Fig. 4B). In Clusters 1–4, which had a low disease burden, there were many significant loci, genes, and gene sets (Table 2) that were mostly linked to the immune system, including major histocompatibility complex genes (*HLA* genes), receptors (interleukin- and Toll-like receptors), and cytokines. This was also reflected in functional enrichment analyses that identified significant enrichment in several gene sets (Supplementary Fig. S11), including *positive regulation of immune system process, regulation of cytokine production, MHC class II protein complex assembly, Toll-like receptor binding, immune receptor activity, cytokine-cytokine receptor interaction,* and *Th1 and Th2 cell differentiation,* involved in the immune response. Genome-wide significant SNPs in Clusters 1–4 exhibited substantial overlap with allergic diseases (asthma, allergic rhinitis, eczema, and hay fever), cardiometabolic traits (BMI, C-reactive protein [CRP], and high-density lipoprotein [HDL]), chronic diseases (rheumatoid arthritis, multiple sclerosis, and diabetes), inflammatory conditions of the colon (inflammatory bowel disease), and blood measures (white blood cell count and vitamin D level), consistent with results from gene-based and gene set-based analyses, which showed a strong link to immune-related biological processes (Supplementary

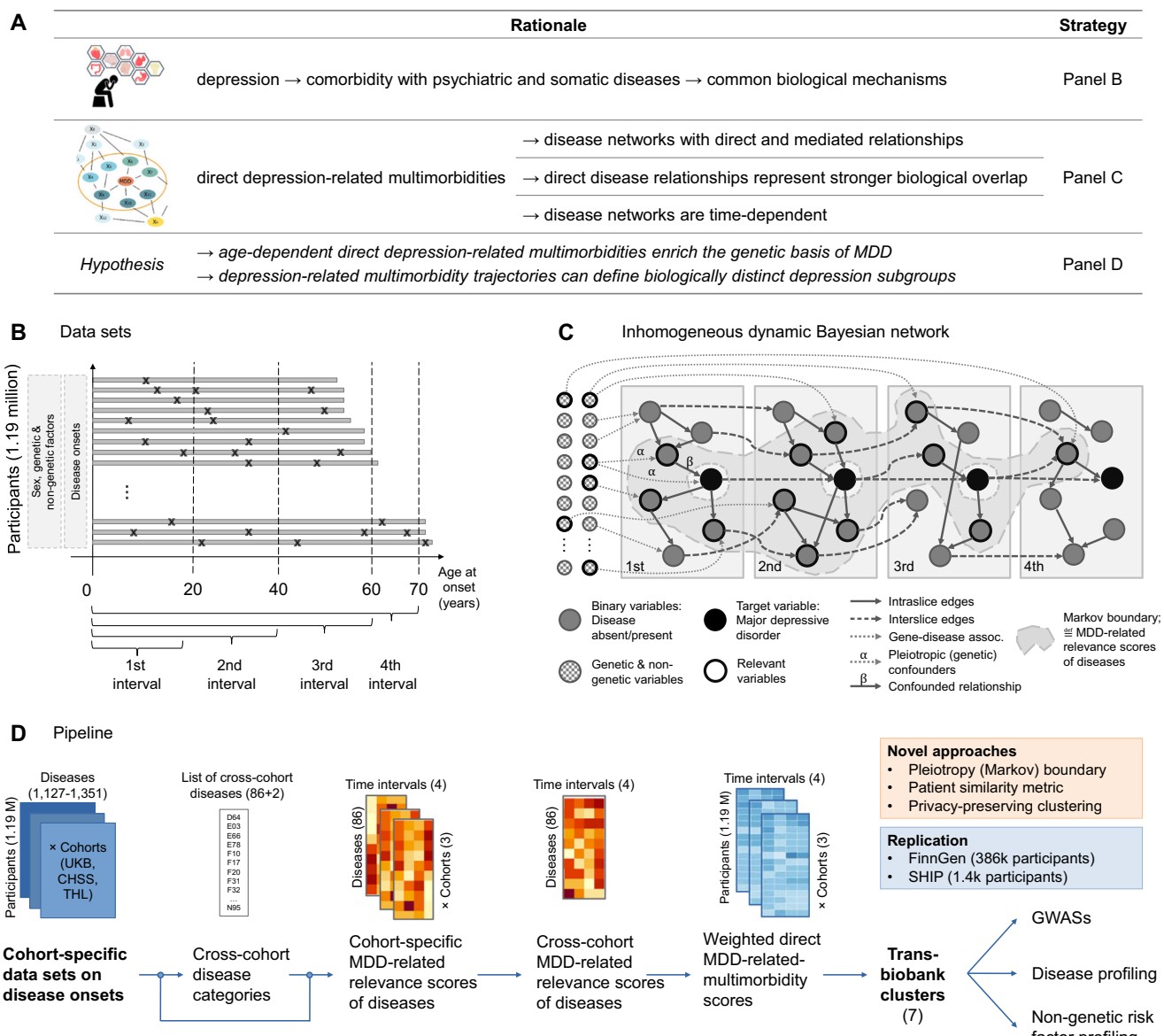

**Fig. 1 | Overview of the main methods. A** Rationale and hypothesis of the study. Accumulating evidence suggested that MDD is frequently comorbid not only with other psychiatric disorders but also with several somatic diseases contributing to worse health-related outcomes and decreasing quality of life[68–71]. Thanks to network medicine and system biology approaches, it has been demonstrated that comorbid conditions partially represent common biological mechanisms[72–75]. Furthermore, directly related comorbidities of depression, where the relationships are not mediated by other disorders, represent stronger molecular-level relationships[15] and are time-dependent (i.e., vary with onset age[76]). Finally, a recent comorbidity mapping study of asthma supported that comorbidities are indeed suitable to delineate distinct subgroups of complex multifactorial disorders[77]. **B** The cohort-specific datasets contain the onset ages of diseases in three-character ICD-10 categories. Data were collected from the participants over various periods, depicted by the length of the grey lines, with disease onsets marked by an 'x'. Participant trajectories were discretized into cumulative time intervals, as shown at the bottom of the figure. **C** The structure of the inhomogeneous dynamic Bayesian network

used. The boxes correspond to intervals, the nodes in the boxes correspond to diseases, and the solid and dashed edges indicate direct relations between the diseases. This method determined the *strongly relevant* MDD-related multimorbidities; these nodes are in the Markov boundary of the target variable, indicated by the grey-shaded region and a thick black node border. Genetic and other non-genetic variables also influenced the onset of the diseases (dotted edges). One aim of the study was to identify pleiotropic genetic variants (edges with α) that influence the onset of MDD and its related multimorbidities. These variants confound the direct relationship (edge β) between MDD and its strongly relevant comorbid conditions. **D** Overview of the study pipeline. We determined MDD-related cross-cohort clusters of all participants in the UKB, CHSS, and THL cohorts by utilizing the temporal trajectories of the participants' MDD-related multimorbidity burden. The seven identified clusters were then characterized based on disease and non-genetic risk-factor profiles and genetic contributions, and the findings were validated in the two independent cohorts (the FinnGen and SHIP cohorts).

Data 18 and 19). In addition, the genetic correlations among these clusters were strong (Fig. 4B).

The three clusters with a high disease burden, Clusters 5–7, exhibited distinct individual patterns. GWAS signals for Clusters 5–6 were weaker, with only a few GWAS loci showing overlapping results with psoriasis (Cluster 5) and cardiovascular conditions, asthma, rheumatoid arthritis and different blood measures (Cluster 6)

(Table 2). In contrast, Cluster 7 had a negative genetic correlation with all previous clusters and the strongest genetic contribution. Thus, the effect alleles of significant SNPs completely differed between low-disease-burden clusters and Cluster 7. Due to the high number of identified loci, overlap with other diseases was high and included apolipoprotein A1, coeliac disease, coronary artery disease, vasculitis, and cholangitis (in addition to the diseases associated with the other

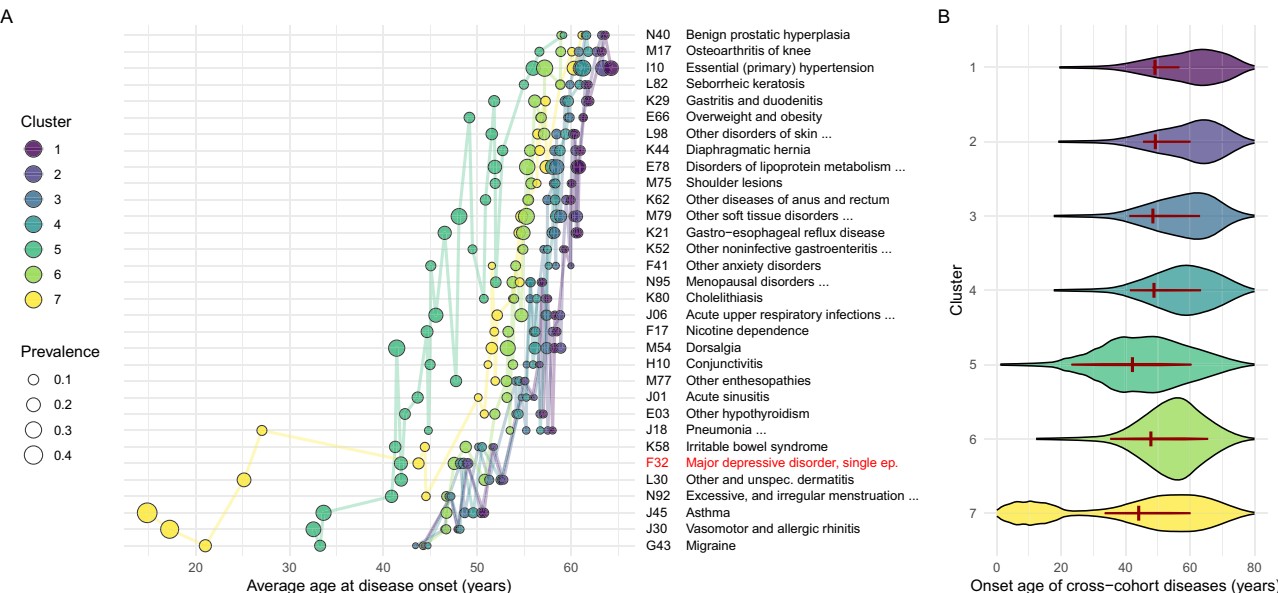

**Fig. 2 | Temporal disease patterns in the clusters according to UKB** (**N** = 502,504). **A** The average onset ages of the most prevalent (>5%) cross-cohort diseases (per line) according to the seven clusters in the UKB cohort. The node colour indicates the clusters, and the node size is proportional to the observed prevalence of the disease (in %) in the cluster. MDD is highlighted with red colour. **B** The onset distribution of all cross-cohort diseases in the clusters of the UKB

cohort is shown as violin plots, where the colour indicates the cluster. Each of the seven violin plots represents one cluster, computed using data from the entire cohort (*N* = 502,504), with individuals weighted by their posterior probability of belonging to the respective cluster. The onset distribution and the mean age of MDD onset are indicated with inline red violin plots and red vertical bars, respectively.

clusters). A comparison of significant genes and gene sets among clusters is shown in Fig. 4D and Supplementary Fig. S11. Most genes were shared with Cluster 7, as it contained by far the highest number of significant genes.

To evaluate genetic similarity with other major psychiatric disorders, we assessed genetic correlations with Psychiatric Genomics Consortium (PGC) GWAS results for MDD, bipolar disorder (BD) and schizophrenia[6,17,18]. Only Clusters 5–6 showed a significant positive genetic correlation with MDD, whereas Clusters 1–2 exhibited negative genetic correlations with MDD and BD and Cluster 7 a positive correlation with BD. Regarding schizophrenia, no cluster exhibited a significant genetic correlation. Regarding asthma (GWAS on UKB data), genetic correlations revealed a similar pattern as the genetic correlations among clusters, with negative correlations in Clusters 1–4, positive correlations in Clusters 5&7, and no significant correlation in Cluster 6. Moreover, we performed a UKB-specific case-only analysis focusing on individuals diagnosed with MDD, and found substantial genetic correlations (0.78–1) between the original population-based clusters and the MDD-specific clusters, underscoring a significant genetic similarity across these groups (Supplementary Fig. S12). However, the reduced sample size (*N* = 28,853) in the MDD case-only scenario led to lower heritability estimates compared to the original clusters. Extending the analysis to various depression phenotypes (Supplementary Fig. S12) showed high genetic correlation among these and highly similar genetic correlation patterns observed between them and the clusters.

To compare clinical observations with genetic predispositions, we calculated genetic correlations between the seven MDD-related clusters and all 86 cross-cohort diseases in the UKB cohort. The clinical and genetic correlations were comparable (Supplementary Figs. S13 and S14), and Clusters 1–4 and 7 were mainly associated with lower genetic risk for the diseases ($r_g$ < 0), whereas Clusters 5–6 were associated with a higher disease burden.

Finally, these results allowed us to assess the extent of *pleiotropy* among the clusters and MDD at the level of genes and functional modules of the human interactome, as pleiotropy may point to shared underlying genetic mechanisms between MDD and the clusters. At the gene level, we defined pleiotropy as the intersection of statistically significant genes in each cluster and MDD-associated genes according to the latest GWAS meta-analysis[5]; in brief, we found 17 pleiotropic genes (Table 2). Our results demonstrate significant enrichment of MDD-associated genes within the clusters, validating our hypothesis that strongly relevant MDD-related multimorbidities enhance the genetic background of MDD. According to the hypergeometric test, significant overlap was observed between MDD genes and cluster-specific genes in three clusters. Additionally, gene set enrichment analysis revealed that MDD genes were consistently and significantly enriched across the ranked list of cluster-specific genes in all seven clusters. Moreover, we identified cluster-specific functional modules significantly influenced by MDD-associated genes; thus, they can be considered pleiotropic. We identified 31 relevant modules, at least one in each cluster (Supplementary Fig. S15.1–S15.7), which indicates that network-based enrichment captured greater pleiotropy between MDD and the clusters at the level of functional modules than at the level of individual genes. In these modules, several other MDD-associated genes had a significant pleiotropic influence, such as *ETFDH*, *PAX5*, *ZDHHC5*, *DENND1B*, *PLCG1*, *MICB*, *STK19*, *CDK14*, *EP3OO*, *ERBB4*, *RERE*, *BAG5*, *CNTNAP5*, *LRP1B*, *NRG1*, *POGZ*, and *XRCC3*. These findings could provide insights into the complex time- and comorbidity-dependent courses of MDD, which may guide the development of novel long-term therapeutic and pharmaceutical approaches.

## Non-genetic risk-factor profiles of MDD-related multimorbidity clusters in the UKB cohort

To assess the non-genetic risk factors collected cross-sectionally within the clusters, we examined associations of behavioural and physiological factors with the MDD-related clusters. This analysis determined the specific risk-factor profiles for each identified MDD-related cluster, offering a snapshot of all participants at the time when these factors were evaluated. Clusters 1–4 and Clusters 5–6 were similar as a group but considerably different from each other. Cluster 7 appeared unique in terms of several factors (Fig. 5A, Supplementary

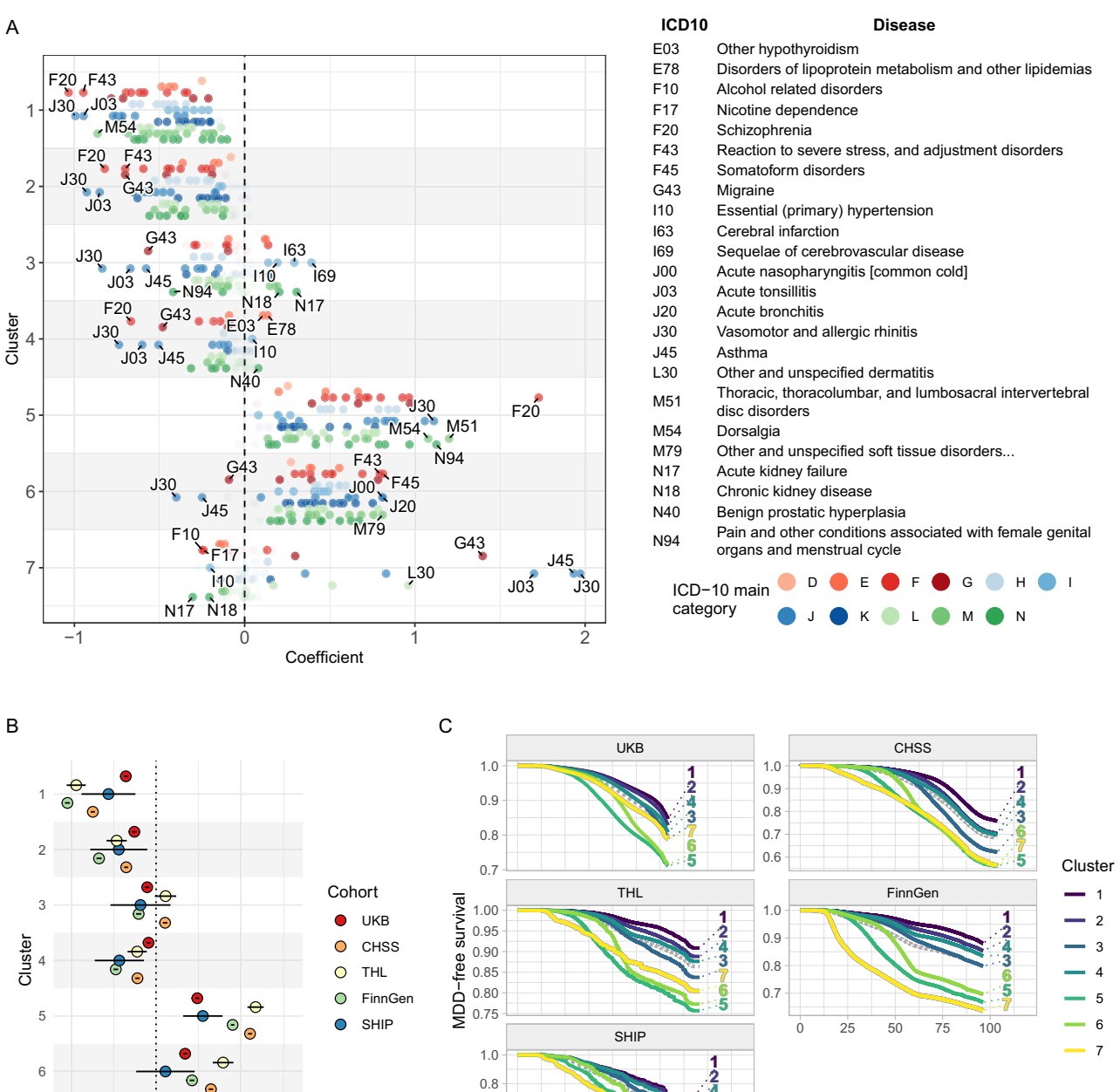

| ICD10 | Disease |
|---|---|
| E03 | Other hypothyroidism |
| E78 | Disorders of lipoprotein metabolism and other lipidemias |
| F10 | Alcohol related disorders |
| F17 | Nicotine dependence |
| F20 | Schizophrenia |
| F43 | Reaction to severe stress, and adjustment disorders |
| F45 | Somatoform disorders |
| G43 | Migraine |
| I10 | Essential (primary) hypertension |
| I63 | Cerebral infarction |
| I69 | Sequelae of cerebrovascular disease |
| J00 | Acute nasopharyngitis [common cold] |
| J03 | Acute tonsillitis |
| J20 | Acute bronchitis |
| J30 | Vasomotor and allergic rhinitis |
| J45 | Asthma |
| L30 | Other and unspecified dermatitis |
| M51 | Thoracic, thoracolumbar, and lumbosacral intervertebral disc disorders |
| M54 | Dorsalgia |
| M79 | Other and unspecified soft tissue disorders... |
| N17 | Acute kidney failure |
| N18 | Chronic kidney disease |
| N40 | Benign prostatic hyperplasia |
| N94 | Pain and other conditions associated with female genital organs and menstrual cycle |

**Fig. 3 | Disease risk in the clusters of the discovery and validation cohorts.**
**A** Coefficient of cluster membership (hazard ratio [HR] in the weighted Cox proportional hazards regression model) with respect to the onset of each cross-cohort disease in the UKB cohort ($N = 502{,}504$). The top five diseases with the strongest increase/decrease in risk in each cluster are indicated in the plot and listed on the right. The colour of the markers corresponds to the main ICD-10 disease category. D: Diseases of the blood and blood-forming organs, E: Endocrine, nutritional and metabolic diseases, F: Mental, behavioural and neurodevelopmental disorders, G: Diseases of the nervous system, H: Diseases of the eye, ear and mastoid process, I: Diseases of the circulatory system, J: Diseases of the respiratory system, K: Diseases of the digestive system, L: Diseases of the skin and subcutaneous tissue, M: Diseases

of the musculoskeletal system and connective tissue, N: Diseases of the genitourinary system. **B** Values and 95% confidence intervals of cluster membership coefficients (hazard ratios) from the weighted Cox proportional hazards regression models for the onset of MDD across various cohorts. Points indicate coefficient values and error bars represent the 95% confidence intervals. Colours represent the different cohorts. **C** Weighted Kaplan–Meier estimates of MDD-free survival in the various cohorts throughout participants' lifespans. Survival curves are labelled by cluster numbers, and the colours of the curves indicate the distinct clusters. The dotted grey curves indicate the mean MDD-free survival in the whole cohort, regardless of cluster membership. (In **A** and **C**: UKB $N = 502{,}504$; THL $N = 41{,}092$; CHSS $N = 645{,}913$; FinnGen $N = 385{,}640$; SHIP $N = 1449$).

Fig. S16). The structure of the non-genetic risk-factor profiles of the clusters was consistent with the observations made at the clinical and genetic levels in the UKB cohort. In sum, Clusters 1–2 were associated with a higher age but a lower burden of several behavioural risk factors.

In contrast, Clusters 3–4 were associated with increases in several behavioural and physiological risk factors, including higher BMI, lower education level and smoking, as well as lower income and insomnia (in Cluster 4). In contrast, the three remaining clusters were associated

**Table 2 | Summary of the GWAS analysis of the MDD-related clusters in the UKB cohort (N = 249,167)**

| | Cluster 1 | Cluster 2 | Cluster 3 | Cluster 4 | Cluster 5 | Cluster 6 | Cluster 7 |
|---|---|---|---|---|---|---|---|
| *N* per SNP | 246,639–249,167 | | | | | | |
| λ | 1.2168 | 1.1523 | 1.0772 | 1.0864 | 1.0833 | 1.1843 | 1.1555 |
| *h²* | 0.0483 | 0.0331 | 0.0171 | 0.0186 | 0.0148 | 0.037 | 0.0385 |
| SNPs | 6,266,283 | | | | | | |
| Significant SNPs[a] | 1908 | 1952 | 1451 | 1924 | 1 | 52 | 5986 |
| Significant loci[a] | 15 | 18 | 14 | 13 | 1 | 3 | 36 |
| Significant genes[b] | 87 | 62 | 51 | 110 | 5 | 15 | 271 |
| Significant gene sets[c] | 62 | 67 | 72 | 87 | 3 | 0 | 129 |
| Significant MDD-associated genes[d] | *HLA-DQA1, HLA-DQB1, LIN28B, LST1, HLA-B* | *HLA-DQA1, HLA-DQB1, ITPR3, HLA-B* | *HLA-DQA1, HLA-DQB1, HLA-B* | *HLA-DQA1, HLA-DQB1, HLA-B, PSORS1C1, PSORS1C2* | *LST1* | *PSORS1C1, PSORS1C2* | *HLA-DQA1, HLA-DQB1, HLA-B, HSPA1A, HSPE1-MOB4, LST1, MOB4, PLCL1, PSORS1C1, PSORS1C2, RFTN2, SLC44A4, SPPL3, SF3B1, TRAF3* |
| Enrichment of MDD-associated genes within cluster-specific genes | $P\text{-value}_{HGT} = 0.037$ $P\text{-value}_{GSEA} < 10^{-16}$ | $P\text{-value}_{HGT} = 0.047$ $P\text{-value}_{GSEA} = 5.44 \times 10^{-7}$ | $P\text{-value}_{HGT} = 0.064$ $P\text{-value}_{GSEA} = 6.55 \times 10^{-7}$ | $P\text{-value}_{HGT} = 0.064$ $P\text{-value}_{GSEA} = 1.56 \times 10^{-10}$ | $P\text{-value}_{HGT} = 0.065$ $P\text{-value}_{GSEA} = 3.79 \times 10^{-5}$ | $P\text{-value}_{HGT} = 0.064$ $P\text{-value}_{GSEA} < 10^{-16}$ | $P\text{-value}_{HGT} = 2.77e\text{-}05$ $P\text{-value}_{GSEA} = 6.55 \times 10^{-7}$ |

λ: genomic inflation factor based on LD score regression; $h^2$: genetic heritability.

Association analyses were performed for each cluster using linear regression to test the association between each SNP and the posterior log odds of cluster membership, controlling for age, sex, the first ten genetic principal components, and the genotyping array. The significance of the over-enrichment of MDD-associated genes (according to Howard et al. [5]) within the cluster-associated genes was assessed by conducting one-sided hypergeometric tests and Gene Set Enrichment Analysis (GSEA). To account for multiple comparisons, we applied Holm's correction method.

*GSEA* Gene Set Enrichment Analysis, *HGT* Hypergeometric test.

[a]The significance of SNPs refers to a genome-wide significance level of $5 \times 10^{-8}$.

[b]Results from MAGMA gene-level analyses, significance based on Holm correction, using 19,843 protein-coding genes.

[c]Results from the g:Profiler method, significance based on the g:SCS correction algorithm at a significance threshold of 0.01.

[d]Overlapping significant MDD-associated genes according to the GWAS meta-analysis by Howard et al. [5] of a total of 269 genes.

with lower age; Clusters 5–6 were additionally associated with overall increased risk, including stress and psychological traits, while Cluster 7 had a more favourable risk-factor profile in general.

**Validation of MDD-related multimorbidity profiles at the genetic and non-genetic risk-factor levels**

In the additional cohorts, we validated the characteristics of MDD-related multimorbidity profiles identified in the UKB cohort on all levels. Validation of genetic findings was performed in the Finnish cohorts (FinnGen and THL cohorts). These isolated populations are of special interest as they are more likely to exhibit deleterious variants and yield previously unknown genetic associations[19]. Although the THL cohorts were considerably smaller, the overall pattern of correlations among the clusters was replicated in GWAS analyses (Supplementary Fig. S17) and comparable to those of the UKB cohort. A GWAS of the 23,786 participants, including 8,711,904 SNPs, revealed very small or negative heritability estimates, indicating limited power to detect significant loci in this cohort[20]. As replication at the individual genetic level was not feasible, we conducted validation at the level of aggregated genetic signals using polygenic risk scores (PRSs) derived from the UKB summary statistics. Thus, all PRSs showed a significant positive association with the cluster probability in the THL cohorts (Benjamini–Hochberg adjusted *p*-values range from $1.0 \times 10^{-15}$ to $1.7 \times 10^{-2}$, Supplementary Fig. S18, see Supplementary Data 20 for details of explained variance by the PRS).

Using the data from the FinnGen cohort (*N* = 277,252; 9,706,223 SNPs), which had a sample size comparable to that of the UKB cohort, a large proportion of genetic findings were replicated at the levels of SNPs, genes, and functional enrichment (Table 3; Supplementary Figs. S19.1–S20, S11; Supplementary Data 21–30). The replicated genes included *HLA* genes, especially in Clusters 1–2, and several additional genes throughout the genome related to Clusters 1–4 and 7 (Supplementary Fig. S20; Supplementary Data 21B). Genetic correlation patterns among the clusters largely overlapped with the patterns observed in the UKB cohort (Fig. 4C), and the strong genetic correlations among clusters in the FinnGen and UKB cohorts pointed to similar genetic factors driving the associations (Supplementary Data S31). Significant loci in the FinnGen clusters also revealed a strong link with immunological phenotypes (rheumatoid arthritis, asthma, and IgG levels). The functional enrichment analysis showed several Gene Ontology (GO) terms that overlapped with the UKB cohort, such as MHC protein complex assembly and MHC class II receptor activity, as well as overlaps with numerous Kyoto Encyclopedia of Genes and Genomes (KEGG) pathways related to various diseases, such as type I diabetes mellitus, rheumatoid arthritis, asthma and IBD in Clusters 1–2 (Supplementary Fig. S11).

The reliability of the non-genetic mostly behavioural risk factors was validated in the THL cohorts (Supplementary Data 1). Although overall effects were weaker due to the smaller sample size, the pattern was similar to that in the UKB cohort (Fig. 5B, Supplementary Fig. S21). In general, Clusters 1–4 had more protective factors, whereas a greater accumulation of risk factors was observed in Clusters 5–6, and Cluster 7 contained a mixture of protective factors and risk factors. Collectively, these findings show that the characteristics of the MDD-related multimorbidity clusters, including genetic contributions and non-genetic risk factors, were validated in additional cohorts, supporting the robustness and generalizability of the results.

As only a few cohorts had a lifetime assessment of disease onset ages, the applicability of the clusters was tested in a setting with limited availability of disease information. The SHIP study only provided information for 13 cross-cohort diseases (Supplementary Data 2) to generate the seven MDD-related comorbidity clusters, as described above. The correlations among the derived cluster probabilities exhibited a pattern similar to that in the larger UKB and FinnGen cohorts (Supplementary Fig. S7). Simulations also showed that the SHIP dataset had an accuracy of 67.5%, which was far better than that expected with a totally random null model (14%) and halfway between the randomly chosen variable set of the same size (52%) and the optimal set of the same size (81%) (Supplementary Fig. S22).

To evaluate the biological meaningfulness of the profiles, we calculated PRSs, similar to the THL cohorts, and assessed the profiles of non-genetic risk factors in comparison with those of the UKB cohort.

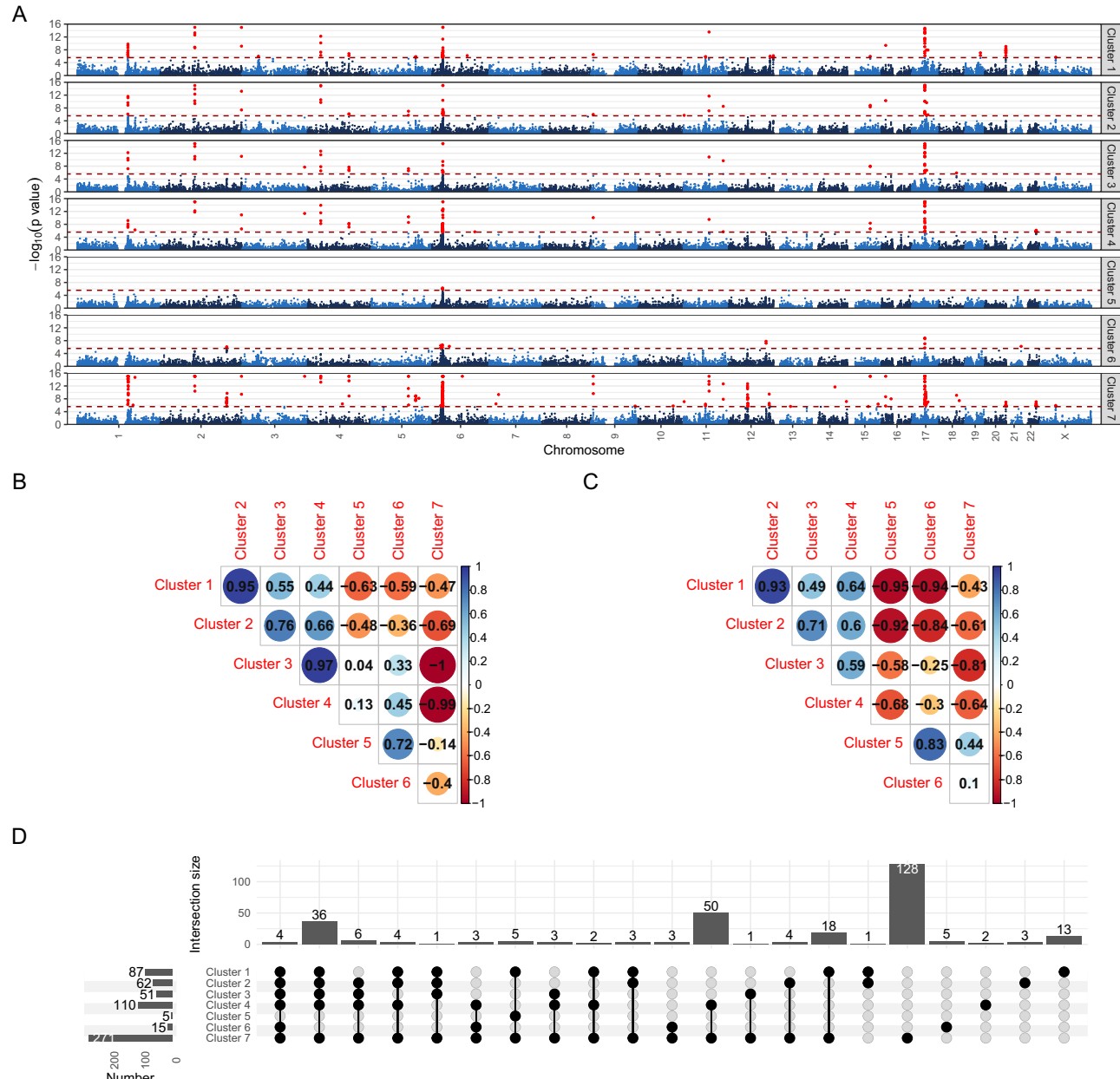

**Fig. 4 | Results from GWAS analyses in the UKB (*N* = 249,167) and FinnGen cohorts (*N* = 277,252).** **A** Gene-based genome-wide Manhattan plots for the seven clusters in the UKB cohort. Association analyses were first performed for each cluster using linear regression to test the association between each SNP and the posterior log odds of cluster membership, controlling for age, sex, the first ten genetic principal components, and the genotyping array. Next, MAGMA gene-level analysis was performed to identify putative significant genes using a SNPwise-multi model, defining the SNP set of each gene with a ± 10 kb window. In the plot, nominal *p*-values are displayed. The genome-wide significant genes are indicated with red dots, and the significance threshold ($2.7 \times 10^{-6}$) is depicted with a dashed dark red

line. **B** Genetic correlation ($r_g$) plot from GWAS summary statistics on the posterior log-odds of cluster membership among Clusters 1–7 in the UKB cohort. The colour of the dots indicates the value of the genetic correlation. **C** Genetic correlation ($r_g$) plot from GWAS summary statistics on the posterior log-odds of cluster membership among Clusters 1–7 in the FinnGen cohort. The area and the colour of the circles represent the magnitude and direction (blue = positive, red = negative) of the genetic correlation between two clusters. **D** Overlap between genome-wide significant genes from MAGMA analyses of Clusters 1–7 in the UKB cohort. Black dots indicate clusters within the comparison. The intersection size corresponds to the number of genes uniquely shared by these clusters.

Five MDD-related cluster PRSs in the SHIP cohort (*N* = 1108) were positively correlated with their cluster membership probability; for Clusters 1 and 7 these correlations reached (suggestive) significance ($p_{cl1} = 0.025$; $p_{cl7} = 0.067$, see Supplementary Data 20 for details of explained variance by the PRS). The correlation pattern among these seven PRSs was similar to the correlation patterns observed at the phenotypic and genetic levels in the UKB and FinnGen cohorts (Supplementary Fig. S23). At the level of non-genetic factors, the association patterns of clusters with age, BMI, blood pressure, insomnia,

neuroticism score, and current depression were replicated in the SHIP cohort (Supplementary Fig. S24). These findings suggest that the MDD-related multimorbidity clusters can be applied to settings with limited disease information, which further supports the generalizability of our approach.

## Discussion

In the TRAJECTOME project, we utilized information on lifetime disease trajectories to define clusters based on temporal patterns of

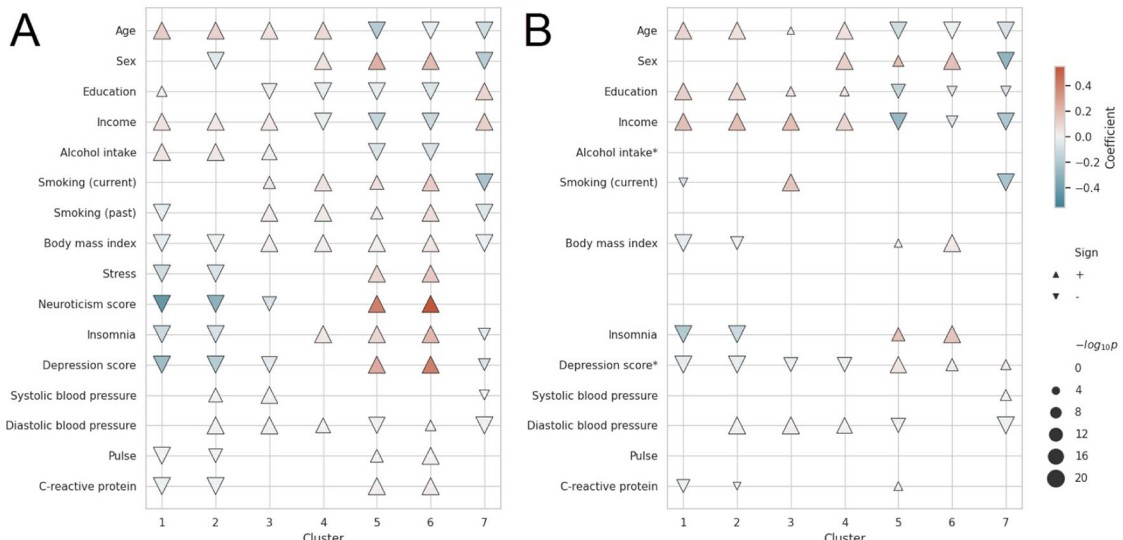

**Fig. 5 | Non-genetic risk-factor profiles for each cluster in the UKB ($N = 249,167$) and THL cohorts ($N = 23,786$).** Simple linear regression models, including one factor at a time with age and sex as covariates, were calculated for each cluster in the **A** UKB cohort and **B** THL cohorts. The posterior log-odds of being in a given cluster were the dependent variable. The direction of the triangles reflects the sign of the coefficient (upwards = positive; downwards = negative), and the colour reflects the magnitude. Statistical analyses were performed using two-sided $t$-tests to assess the significance of each factor's effect on cluster membership. Adjustments for multiple comparisons were made using the Bonferroni correction. The size of the triangles is proportional to the $-\log_{10}$ $p$-value, and only significant values are shown ($-\log_{10}(p) > 4$). Sex was coded as follows: 1 = male, 2 = female. The risk factors of stress and neuroticism score were not available in the THL cohorts. *Alcohol intake and depression score were not available in the FinHealth17 and Finrisk cohorts, respectively, but were available in the other THL cohort.

### Table 3 | Overview of validation results in the FinnGen cohort ($N = 277,252$)

| | MDD-related cluster membership | | | | | | |
|---|---|---|---|---|---|---|---|
| | **Cluster 1** | **Cluster 2** | **Cluster 3** | **Cluster 4** | **Cluster 5** | **Cluster 6** | **Cluster 7** |
| Significant SNPs in the FinnGen cohort[a] | 221 | 128 | 0 | 9 | 0 | 0 | 6 |
| Significant loci in the FinnGen cohort[a] | 9 | 4 | 0 | 1 | 0 | 0 | 1 |
| Significant genes in the FinnGen cohort[b] | 75 | 17 | 3 | 2 | 3 | 13 | 14 |
| UKB GWAS hits available in the FinnGen cohort | 1376 (72%) | 1467 (75%) | 1052 (72%) | 1495 (78%) | 1 (100%) | 48 (92%) | 4950 (83%) |
| UKB GWAS hits available and nominal significant in the FinnGen cohort | 870 | 938 | 759 | 703 | 0 | 43 | 2260 |
| …and same direction of effect (% SNPs of available UKB GWAS hits replicated) | 866 (63%) | 938 (64%) | 759 (72%) | 699 (47%) | 0 (0%) | 43 (90%) | 2254 (46%) |
| Replicated loci[c] | 9 (60%) | 8 (44%) | 4 (29%) | 6 (46%) | 0 (0%) | 2 (67%) | 16 (44%) |
| Replicated genes[d] (% of UKB GWAS gene hits replicated) | 9 (10.3%) | 6 (9.7%) | 1 (2%) | 1 (0.9%) | 0 (0%) | 0 (0%) | 13 (4.8%) |
| $r_{g\_UKB-FG}$ | 0.6521 | 0.5956 | 0.5389 | 0.3693 | 0.7652 | 0.8005 | 0.4521 |

Association analyses were performed for each cluster using linear regression to test the association between each SNP and the posterior log odds of cluster membership, controlling for age, sex, and the first ten genetic principal components.
[a]The significance of SNPs refers to a genome-wide significance level of $5 \times 10^{-8}$.
[b]Results from MAGMA gene-level analyses, significance based on Holm correction, using 19,173 protein-coding genes.
[c]Number of replicated loci at the nominal significance level ($p$-value < 0.05).
[d]Number of replicated genes at the genome-wide significance level (Holm's-adjusted $p$-value < 0.05).

**MDD-related multimorbidity burden.** Based on data from 1.2 million individuals in the general population of three European countries (i.e., discovery cohorts), we identified seven MDD-related clusters with distinct clinical, genetic, and non-genetic risk-factor profiles. We further validated their applicability in independent cohorts, including those with a limited set of disease information. Based on these profiles, we extracted biologically meaningful information that can be used to interpret the clusters with respect to their aetiology, clinical relevance, and possible disease prevention strategies.

We identified four clusters with a low-risk profile accompanied by a low burden of MDD and MDD-related disorders as well as three clusters associated with risk-conferring profiles. Individuals in Clusters 1–4 were healthy until older age with a low risk of developing MDD.

This favourable clinical pattern regarding MDD was also observed at the levels of genetic and non-genetic risk-factor profiles.

With the strong genetic correlations among these clusters, GWAS results suggested a substantial and consistent contribution of the immune system and protection against allergic, autoimmune, and inflammatory diseases. However, clear differences in MDD-related pleiotropic functional genetic modules supported more specificity at the genetic network level. For example, individuals in Clusters 3–4 developed age-related cerebrovascular or metabolic disorders, possibly due to a slight increase in genetic vulnerability as well as less-favourable lifestyle habits. Based on large-scale meta-analyses, depression risk is also dependent on non-genetic behavioural factors, including sleep, stress, diet, physical activity and social interactions[21,22].

Our approach extends these findings by using genetic and clinical data to identify biological subgroups resilient to depression on multiple levels.

In contrast, Clusters 5–6 exhibited a high-risk profile in terms of the prevalence of MDD and most MDD-related disorders. The non-genetic risk-factor profile was largely disadvantageous, and the genetic profile revealed a strong correlation between Clusters 5 and 6, albeit distinct MDD-related pleiotropic genetic functional networks. Differences between these clusters were mainly due to clinical manifestations, with Cluster 5 showing earlier MDD onset and the highest risk of schizophrenia; Cluster 6 showed later MDD onset accompanied by an increased risk of stress-related behavioural problems and somatoform disorders. This pattern of high MDD risk combined with multimorbidities and poor lifestyle habits might contribute to the worst outcomes, as a high multimorbidity burden has a deleterious effect on the clinical course of MDD[23], and the quality of life is dramatically lower in depression patients with chronic physical conditions[24]. A recent analysis confirmed this pattern of known behavioural risk factors for depression using UKB data, proposing that inflammatory processes are a common neurobiological pathway[25]. Thus, our study provides clear evidence of disease and risk-factor patterns related to MDD that might benefit from behavioural interventions.

In contrast to the high disease burden in Clusters 5–6, which were strongly associated with non-genetic risk factors, Cluster 7 showed a strong contribution of inflammation-related genetic predispositions regardless of non-genetic factors. The early manifestation of cluster assignment also suggests a substantial contribution of genetics rather than long-term non-genetic risk factors. Although most MDD-related diseases showed decreased prevalence in this cluster, MDD and a group of respiratory disorders (asthma and allergic rhinitis) exhibited sharply increased prevalence, driving the risk association. A strong link between MDD and highly heritable, usually early-onset immune-related diseases has been identified previously at both phenotypic and genetic levels[26–31]. The strong genetic correlations between these diseases and Clusters 5&7 point to genetic subgroups within depression that have a shared aetiology.

The findings obtained by using our approach confirmed that inflammatory signalling is part of the underlying aetiology of depression[32] in Cluster 7. Despite the low genetic risk of MDD in Clusters 1–4 and Cluster 7 as well as the advantageous behavioural profiles, Cluster 7 had an increased risk for early-onset depression, which might be due to contrasting profiles of genes involved in inflammatory signalling pathways. However, in Cluster 5, besides genetic risk, external stressors (such as psychosocial stress or disadvantageous behavioural factors[33]) may also contribute to inflammation. Thus, the temporal trajectories of MDD-related multimorbidity clusters in our study revealed highly pleiotropic inflammation-related genetic loci that exert protective or risk effects in a cluster-specific manner by engaging distinct molecular networks and non-genetic risk factors. The observed heterogeneity within MDD risk could explain previous contradictory findings regarding the relationship between MDD and inflammatory genes[5,6,34–36]. Our cluster-related findings also corroborate previous inflammatory, metabolic and related hypotheses regarding pathomechanisms of depression[37–40]. At the clinical level, further detailed phenotyping[4] of MDD patients within these clusters could enable phenotypic and biological subtyping of depression to develop targeted prevention and intervention strategies.

Although our method was supported by temporal disease information from public health data, this approach had several limitations and involved simplification. Differences in healthcare systems lead to differences in disease rates, possibly related to differences in year of birth and age. Additionally, our method is currently unable to distinguish between chronic and acute diseases, which may have different long-term impacts on the development of depression and other conditions. However, the cross-cohort acute diseases were mainly acute inflammatory diseases that may share pathophysiology with other immune-related diseases and depression. Finally, our Bayesian network methodology is sensitive to unknown confounders and selection bias, potentially causing spurious correlations, but the fairly complete nature of our cohorts and the cross-cohort design mitigate this danger.

In conclusion, we identified seven MDD-related multimorbidity clusters with unique genetic and non-genetic risk-factor profiles that highlight the involvement of neuroinflammatory processes in depression and provide a strategy for subtyping depression patients. This bridges the gap between complex multimorbidity patterns associated with MDD over the course of an individual's life and recommendations for prevention, early intervention and personalized psychological and pharmacological therapy. This approach could also be expanded to other complex diseases with a high load of comorbidities and shared genetic and non-genetic risk factors.

## Methods

### Description of the training cohorts

**UK Biobank.** Under application number *1602*, we extracted data from the UK Biobank (UKB) database, which includes medical and phenotypic data of participants recruited from NHS patient registers of people aged 40–69 years[41]. Ethical approval was given by the National Research Ethics Service Committee North West–Haydock[42], and all participants provided written informed consent. All procedures were conducted in accordance with the Declaration of Helsinki.

To identify MDD-related clusters based on disease trajectories, 502,504 participants who had available disease onset information for 1,127 ICD-10 categories were included (for details, see the "Cross-cohort disease categories and relevance scores" section).

**Quality control of GWAS data in UKB data.** Our genomic quality control (QC) methods were detailed previously[43], but in the present analyses, we did not restrict participants according to the availability of complete dietary data.

Specifically, we selected participants with White British ancestry (UKB data field 22006, defined both by self-report and genetic ancestry) and without putative sex chromosome aneuploidy (data field 22019). We used v3 genetic data of UKB with genotyped variants and, when genotyped variants were not present, we used imputed variants as well and positioned them according to the GRCh37/hg19 genome assembly. Variant QC consisted of several steps. First, multiallelic variants as well as variants with a minor allele frequency (MAF) < 0.01 were excluded, retaining only single-nucleotide polymorphisms (SNPs). For imputed SNPs, both info and certainty parameters had to be at least 0.9. Furthermore, SNPs and participants were excluded according to the missingness rate (in an iterative manner, with cut-off points at 0.1, 0.05 and 0.01), and SNPs violating Hardy-Weinberg equilibrium ($p < 1 \times 10^{-5}$) were excluded. Before further calculations for participant filtering, linkage disequilibrium (LD) pruning, with an $R^2$ of 0.2, was applied to SNPs. The maximal set of unrelated individuals was selected[44] (data field 22020), and a kin-cut-off of 0.044 was applied. Finally, a sex check and heterozygosity outlier detection were performed, as described previously[45].

For the GWAS analyses, we selected participants who did not withdraw their consent before February 2022 and did not have missing data on sex, age, or genotyping array. To control for population stratification, principal component analysis was performed with the final set of participants and with the SNP subset after the LD pruning (described above).

**Catalan Health Surveillance System.** Since 2011, the Catalan Health Surveillance System (CHSS) has collected detailed information on healthcare utilization from the entire population of Catalonia

(northeastern Spain; 7.5 million inhabitants). The CHSS integrates the information contained in the Minimum Basic Dataset for Healthcare Units registry provided by 63 hospitals, 49 mental health centres, 370 primary care teams and 72 long-term care centres every 6 months. The CHSS assembles information on the use of public healthcare resources, pharmacological treatments, socioeconomic and educational status, psychological health and other billable healthcare costs, such as nonurgent medical transportation, ambulatory rehabilitation, domiciliary oxygen therapy, and dialysis[46].

From the 7.5 million individuals documented in the CHSS, we considered only registry data from all citizens living in the integrated health district of Barcelona-Esquerra ("AISBE") between 2011 and 2019 ($N = 645{,}913$), with a mainly White European ancestry, as input to identify MDD-related clusters, extracting over 42 million diagnostic codes recorded between 1913 and 2019. Notably, approximately 50% of the records were from the period after 2012, when the Catalan health system underwent digitization and implemented electronic medical records. Conversely, only 2 million records were available from before 2000, with approximately 200,000 records available from before 1950.

**Finnish Institute for Health and Welfare (THL).** For cluster analysis, data from 41,092 participants in Finnish population surveys were included[47]; these surveys included Finrisk 1992 ($N = 5019$), Finrisk 1997 ($N = 7130$), Finrisk 2002 ($N = 7207$), Finrisk 2007 ($N = 4635$), Finrisk 2012 ($N = 5396$)[48], Health 2000/2011 ($N = 6004$) and FinHealth[49] 2017 ($N = 5074$) (https://urn.fi/URN:ISBN:978-952-343-449-3). After excluding related individuals (IBD > 0.2), 30,961 participants were retained from the Finnish population surveys[47] as follows: Finrisk 1997 ($N = 6723$), Finrisk 2002 ($N = 5698$), Finrisk 2007 ($N = 4635$), Finrisk 2012 ($N = 3078$)[48], Health 2000/2011 ($N = 5944$) and FinHealth 2017 ($N = 4883$). These participants, aged 20–100 years, were chosen at random from the Finnish population and represented different parts of Finland. For GWAS data used from THL cohorts, see the "Quality control of GWAS data from the FinnGen and THL cohorts" section.

**Description of validation cohorts**
**FinnGen project.** We used data from the FinnGen project[19] (https://www.finngen.fi/en) from data freeze 10 (DF10; excluding THL cohorts) to generate the MDD-related clusters in an independent cohort from the Finnish population. In brief, FinnGen is a public–private project aiming to collect genotype data from half a million Finnish people and combine these data with data from various health registries. The participants consist of legacy individuals recruited before the start of the FinnGen project and prospective individuals; these latter participants were recruited on a voluntary basis during hospital visits if the patient provided consent for their data to be entered in the biobank.

The THL cohorts and FinnGen cohort contain disease information collected in the following registries: Causes of death (STAT, 1969), Register of Primary Health Care Visits, HILMO (2011), Care Register for Health Care inpatient visits, HILMO (THL, 1969), Care Register for Health Care, specialist outpatient visits, HILMO (THL, 1998), Finnish Cancer Registry (CANC, 1953), Cervical cancer screening (THL, 1991), Breast cancer screening (THL, 1992), and the Finnish Registry for Kidney Diseases (1964).

In total, FinnGen DF10 consists of 430,897 participants with genotype data; after excluding the THL cohorts, 385,640 participants remained for cluster analysis.

**Quality control of GWAS data from the FinnGen and THL cohorts.** The genotyping of FinnGen participants was performed on a Thermo Fisher axiom custom array consisting of 736,145 probes for 655,793 genetic markers. Processing of samples included removing samples where the genetic sex did not match the participant-reported sex in the registries, samples with missing variant information >0.02, samples with excess heterozygosity in common variants (allele frequency >0.05) and samples with excess relatedness to other samples (IBD > 0.1). The processing of variants depended on whether the variant was used in imputation. Quality control included removing variants if allele frequency in the panel was <0.001 (for imputation QC only), removing variants where the allele frequency differed significantly among panels, removing a variant from all batches (FinnGen chip data and legacy data, processed separately) if HWE $p < 10^{-10}$ across all batches; removing any batch missing >0.03 of data (for legacy samples, the missingness threshold was 0.05 due to the exclusion of too many variants for imputation purposes), and removing batches where more than 15% of the batches had missingness >0.04. Finally, variants within a batch were removed if the p-value for HWE was >$10^{-6}$ or if the missingness rate was >0.02. The legacy participants were genotyped on various generations of Illumina GWAS arrays. The Sisuv4 reference panel was used to impute an additional 20,175,454 genetic markers. Information on the generation of the imputation panel and the QC steps used to produce the imputed genotypes is available elsewhere (https://finngen.gitbook.io/finngen-analyst-handbook/finngen-data-specifics/genotype-data/imputation-panel/sisu-v4-reference-panel).

Finrisk cohorts were genotyped at the Sanger Institute, Hinxton, UK; FIMM, Helsinki, Finland and Broad Institute, Cambridge, MA, USA using Illumina HumanCoreExome-12v1, Illumina Human-CoreExome-24v1, Illumina HumanOmniExpress-12v1, Illumina Human610-Quadv1, and Illumina GSAMD-24v1-0_20011747_A1 arrays. The Health 2000/2011 cohorts were genotyped at the Sanger Institute, Hinxton, UK; FIMM, Helsinki, Finland and Broad Institute, Cambridge, MA, USA using IlluminaHuman610K and Human610-Quadv1; Illumina HumanCoreExome-24-v1; and Broad_GWAS_supplemental_15061359_A1 genotyping arrays, respectively. The FinHealth 2017 cohort was genotyped at Thermo Fisher Scientific, San Diego, CA, USA, using the Affymetrix Axiom FinnGen1 array. Before imputation, variants with call rate <0.98, HWE $p < 10^{-6}$, and minor allele count (MAC) < 3 were removed. For THL cohorts, the Sisuv3 reference panel was used to impute an additional 20,175,454 genetic markers. Information on the generation of the imputation panel and the QC steps to produce the imputed genotypes is available elsewhere (https://finngen.gitbook.io/finngen-analyst-handbook/finngen-data-specifics/genotype-data/imputation-panel/sisu-v3-reference-panel).

**Study of Health in Pomerania.** The Study of Health in Pomerania (SHIP) is a general-population-based research project on adult residents in northeastern Germany[50]. In this study, we analysed data from the SHIP-START cohort; in this cohort, at baseline, 4308 White European participants were recruited between 1997 and 2001. To date, three regular follow-ups have been carried out (SHIP-START-1/2/3) as well as a detailed assessment of life events and mental disorders (SHIP-LEGEND) from 2007 to 2010, including 2400 participants from the baseline SHIP-START-0 cohort[51].

For cluster analysis, 1449 participants who took part in SHIP-START-3 and SHIP-LEGEND and had available baseline information from SHIP-START-0 were included. Regarding the age of onset of chronic diseases (Supplementary Data 2), self-reported results were used from baseline and follow-up data. The age of onset for diseases in the F section of ICD-10 codes (F32, F33, F17, F41, F43, F40, and F45) was determined from a combination of self-reported diagnoses from SHIP-LEGEND data and data from the health insurance system that have been collected since the end of 2003. Finally, information for 37 diseases was available. Hereafter, data from the SHIP-START cohort are referred to as SHIP data.

**Quality control of GWAS data in the SHIP cohort.** The SHIP-START-0 participants were genotyped using the Affymetrix Genome-Wide Human SNP Array 6.0. Hybridization of genomic DNA was performed in accordance with the manufacturer's standard recommendations. Genetic data were stored using the database Caché (InterSystems). Genotypes were determined using the Birdseed2 clustering algorithm. For QC purposes, several control samples were added. At the chip level, only participants with a genotyping rate on QC probe sets (QC call rate) of at least 86% were included. Finally, all arrays had a sample call rate >92%. The overall genotyping efficiency was 98.55%. Imputation of genotypes was performed using the HRCv1.1 reference panel and the Eagle and minimac3 software implemented in the Michigan Imputation Server for prephasing and imputation, respectively. SNPs with an HWE $p < 0.0001$ or a call rate <0.95 as well as monomorphic SNPs were removed before imputation.

### Ethics statements

This study, including both the data collection and the current analyses, has received ethical approval from appropriate institutional review boards for all involved cohorts. Specifically, the analysis involved data from the following cohorts: UK Biobank (UKB), Catalan Health Surveillance System (CHSS), Finnish Institute for Health and Welfare (THL), FinnGen, and Study of Health in Pomerania (SHIP). Comprehensive ethical approvals were obtained for each of these cohorts, ensuring that all procedures followed were in accordance with the ethical standards of the responsible committee and with the Helsinki Declaration.

Furthermore, all participants in the study provided written informed consent. Detailed information regarding the ethical approvals, including the specific committees and approval numbers, is available in the Supplementary Information.

### Identification of MDD-related clusters based on disease trajectories

**Assessing diseases strongly relevant to MDD.** We used a Bayesian network-based Markov Chain Monte Carlo (BN-MCMC) method to assess the *strongly relevant variables* with respect to our target variable (MDD). Bayesian networks (BNs) use directed acyclic graphs (DAGs) to represent multivariate dependencies and conditional independencies among the variables. The nodes in these graphs represent variables, and the edges represent direct relationships between the corresponding nodes. Assessments of the complex structure of the variables are called *learning the structure* of the BN based on the observed data. However, in most cases, there are many DAGs with nonnegligible *a posteriori* probabilities (i.e., the best network has many alternatives that are almost as probable as the best network). Even in these cases, there are usually certain structural features, such as the strong relevance of two variables, which can be extracted reliably.

Strongly relevant variables statistically isolate the target variable from all other variables. Therefore, strong relevance is a different concept than a standard pairwise association. First, if the dependency of disease $A$ on the target disease $B$ is indirect (e.g., due to mediation through a third disease $C$), then $A$ and $B$ are associated but not strongly relevant to each other. Second, if $A$ has no direct effect on $B$, but $A$ and $C$ interact with each other to affect $B$ (e.g., disease $A$ does not cause disease $B$, and vice versa, but the presence of diseases $A$ and $B$ together causes $C$), then $A$ is not associated with $B$ but is strongly relevant due to the interactional effect. Therefore, strong relevance indicates either a direct/nonmediated association or an interactional relevance. Below, we refer to a variable's probability of strong relevance with respect to MDD as the variable's *relevance score*.

In the Bayesian learning framework, we can estimate the posterior probability that two variables are strongly relevant to each other (i.e.,

they have a direct influence on each other) as follows:

$$P(\text{strongly-relevant}(X,Y)|D) = \sum_G P(G|D) \cdot I(\text{Edge}_G(X \to Y) \text{ or} \\ \text{Edge}_G(Y \to X) \text{ or } \exists Z : (\text{Edge}_G(Y \to Z) \text{ and } \text{Edge}_G(X \to Z))) \quad (1)$$

where $G$ represents a BN structure (a graph), $D$ is the dataset; $I(.)$ denotes the indicator function, which is 1 if the property holds and 0 otherwise; and $\text{Edge}_G(X \to Y)$ means that an edge points from node $X$ to $Y$ in the $G$ graph. Specifically, the indicator function yields 1 if there is a direct edge from $X$ to $Y$ or from $Y$ to $X$ or if there is a common child node $Z$ of nodes $X$ and $Y$.

Note, that in our methodology, the directed arrows in the Bayesian network represent direct probabilistic relationships between diseases. This means that the presence of one disease (e.g., MDD) directly influences the probability distribution of another. To assess the strong relevance of each disease to MDD, we focused on the concept of the Markov Boundary, which is the smallest set containing all variables carrying information about a target variable that cannot be obtained from any other variable. In other words, we cannot drop any variable from this set without losing information. By examining the diseases that are in the Markov Boundary of the target variable, we calculate the strength of their probabilistic relationship with the target variable. More specifically, within the Bayesian statistical framework employed in our study, we compute the posterior probability of each variable being within the Markov Boundary of MDD (i.e., the probability of their strong relevance with respect to MDD).

We also note that in our Bayesian network framework, we focus on capturing structural probabilistic relationships between variables rather than quantifying interaction terms that occur in regression models. Although these interactions are quantified at the parametric level in Bayesian networks—through the conditional probability distributions of variables given others—our analyses primarily aimed to elucidate the structural relationships by performing exact Bayesian averaging over the parametric level rather than quantifying these interaction effects directly.

It should be also noted that while the relationships in our Bayesian network are direct and unmediated by other diseases, they do not necessarily imply causation. This directness refers to the absence of intermediate variables within the network's model structure, distinguishing these relationships from mere correlations at the abstraction level defined by the entire set of variables in the analysis. However, direct probabilistic relationships in the Bayesian network are derived from observational data, not from interventional studies that manipulate one variable to directly observe its effect on another. Without the ability to control or manipulate the conditions, the relationships might still be influenced by unobserved confounding factors. The direct relationships in the network are based on the strongest statistical dependencies observed in the data, but these dependencies alone do not fulfil all criteria required to establish causality, such as eliminating all potential confounders and demonstrating that the relationship is not reversible.

The posterior probabilities $P(G|D)$ are estimated using a DAG-based MCMC simulation. We applied the Metropolis-coupled Markov Chain sampler with a burn-in period of $2 \times 10^6$ steps and then collected $10^7$ samples (i.e., network structures). We restricted the space of the possible structures by limiting the number of parents per node to 8. Convergence diagnostic testing using Geweke[52] scores indicated that the MCMC chains had converged for 618 out of 621 (99.5%) of the posterior probabilities of the variables' strong relevance, with their z-scores within the acceptable range of −2 to 2, suggesting overall convergence of the chains.

We modelled the participant trajectories using an inhomogeneous dynamic BN to utilize the disease onset information. More specifically, we discretized the first onset time of the diseases to cumulative time intervals ([0–20], [0–40], [0–60], and [0–70]) and

transformed the disease onsets into binary variables that had a value of 1 if the disease was diagnosed in the given time interval and 0 otherwise (Fig. 1B). The dynamic BN accounts for censoring of the participants by including only those participants in each cumulative time interval who have complete disease onset information up to the end of that interval. Then, we separately estimated the strong relevance of all variables with respect to either F32 or F33 (ICD-10 disease categories jointly defining MDD) for each time interval $t$. The variables from time interval $t$-1 were also included in the model, but variables in $t - 1$ could only act as parent nodes, i.e., no edge could point to a variable in $t - 1$. See Fig. 1C for a graphical illustration of this method.

**Cross-cohort disease categories and relevance scores.** As a preliminary step, we determined the set of cross-cohort disease variables as follows. (1) First, for each cohort (UKB, CHSS, and THL), we filtered diseases (according to three-character ICD-10 disease categories) with a prevalence >1% either in the whole cohort or in the subset of depressed participants (i.e., patients diagnosed with either F32 or F33). The primary objective of this pre-filtering step was to exclude rare disorders, as our goal was to identify general multimorbidity trajectory clusters that are broadly applicable. Consequently, this initial filtering led to differing numbers of diseases being considered across each cohort, namely 266 disorders for UKB, 356 for CHSS, and 339 for THL. (2) Next, we estimated the strong relevance of all such diseases with respect to MDD by learning the structural features of the formerly described inhomogeneous dynamic BN. We performed this analysis separately for each cohort. (3) Finally, we selected diseases that had a posterior probability of strong relevance with respect to MDD higher than 0.5 in at least one time interval for at least one cohort, selecting only those disease variables that were consistently available across all cohorts, allowing for uniform analysis across all datasets. This preliminary filtering procedure aimed to gather the broadest possible set of potentially relevant diseases, resulting in 86 *cross-cohort disease categories*. The prevalence rates and summary statistics of the first onsets of these cross-cohort disease categories are shown in Supplementary Data 2 for each cohort.

Finally, we performed the same analyses using only the cross-cohort disease variables together with the sex and household income status variables of the samples. This resulted in cohort-specific relevance scores for each variable, from which we defined *cross-cohort relevance scores* by computing a linear combination of the cohort-specific relevance scores for each time interval by applying uniform weights on the cohorts. The cross-cohort relevance scores are shown in Supplementary Data 3. Computed in this way, the cross-cohort relevance score of a variable corresponds to the expected probability that the variable is strongly relevant with respect to MDD in a given time interval.

**Clustering of participants.** Based on the cross-cohort relevance scores, we computed the *weighted direct MDD-related multimorbidity scores* for each participant in each cohort and for each time interval. The score for the $i$th participant in the $t$th time interval is computed as follows:

$$\text{multimorbidity-score}^{(t)}(i) = \sum_d I(\text{onset}(i, d) \epsilon t) \times \text{relevance-score}^{(t)}(d)$$

(2)

where $d$ represents the cross-cohort diseases, $I(.)$ denotes an indicator function that yields 1 if the first onset of disease $d$ for the $i$-th sample occurs in the $t$-th time interval and 0 otherwise; and *relevance-score$^{(t)}(d)$* denotes the cross-cohort relevance score of disease $d$ in the $t$th time interval. These weighted direct MDD-related multimorbidity scores defined the 4-dimensional space of the samples that we used to cluster the participants.

Finally, we clustered all participants &&using the k-means clustering algorithm in the 4-dimensional space defined by the weighted

direct MDD-related multimorbidity scores. More specifically, the clusters were determined based on the participants with complete observed multimorbidity scores, i.e., participants older than 70 years. In younger participants, one or more multimorbidity scores were not available because there were no observations of their future disease onset. However, based on their partial scores, they were assigned to clusters by allocating them to the cluster with the nearest cluster centre. The number of clusters was determined by manual investigations based on expert knowledge with the help of various cluster metrics (such as the silhouette score of the resulting clusters). See Fig. 1 for a graphical overview of the method and Supplementary Methods for further details on the investigated cluster configurations.

The likelihood of cluster membership for the $i$−th sample and for the $j$−th cluster is defined as:

$$\text{likelihood}_j(i) = \exp\left(-||p_i - c_j||\right)$$

(3)

where $p_i$ and $c_j$ represent the point that correspond to the $i$th sample and the $j$−th cluster's cluster centre, respectively, in the space defined by the multimorbidity scores, and $||p_i - c_j||$ is their Euclidean distance.

The posterior probability of cluster membership for the $i$th sample and the $j$th cluster was the normalized likelihood shown below:

$$P_j(i) = \frac{\exp\left(-||p_i - c_j||\right)}{\sum_k \exp\left(-||p_i - c_k||\right)}$$

(4)

To control for uncertain participant trajectories in the following analyses, we excluded participants for whom the clustering algorithm demonstrated low confidence across all clusters. Specifically, we excluded participants who were both under 60 years of age and whose maximum posterior membership probability did not exceed 0.25 for any cluster. This threshold was chosen to remove individuals for whom the algorithm could not confidently assign a predominant cluster, thereby focusing our analysis on participants with more definitive cluster memberships. This exclusion criterion resulted in a subset of $N$ = 364,008 participants. This subset was used for comparing clusters and deriving age-specific differences and was also the base set for genetic analysis.

In the case of GWAS and non-genetic risk-factor profiling analysis, the posterior log-odds of the cluster memberships were used as target variables as follows:

$$\text{Posterior log odds}_j(i) = \ln \frac{P_j(i)}{1 - P_j(i)}$$

(5)

The posterior probability of cluster membership was used to calculate the disease profile of the clusters.

Our methodology employs a privacy-preserving federated approach to derive the MDD-related clusters across multiple cohorts without sharing individual-level data, making it suitable for collaborative studies where data sharing is restricted. Each participating site independently computes relevance scores for diseases, which are then aggregated to create cross-cohort relevance scores, ensuring that only non-identifiable, summarized information is exchanged between sites. Multimorbidity scores for each participant are calculated by aggregating cross-cohort relevance scores for the diseases they have experienced (see Eq. (2)). These scores are then compiled into counts of occurrences at each site. The final clustering is performed using these aggregated counts, thereby ensuring the confidentiality of individual data throughout the process.

**Software.** Inference over Bayesian network structures was performed with an in-house developed software called BN-BMLA[53]. All other computations were performed in R statistical software (version 4.1.1)[54] or Python (version 3.8). Clusters can be computed with a command

line R script that is available online: https://github.com/gezsi/mdd-clustering.

## Disease profile of MDD-related clusters

We used weighted Cox regression to determine the disease outcomes in the various clusters (i.e., the hazard ratio of cluster membership regarding disease occurrence) independently for each cohort. Specifically, for each cluster, we constructed Cox proportional hazard models, where the independent variable for a specific individual was a dummy variable created in the following way. We counted each participant twice, summing to a weight of 1. First, we set the value of the dummy variable to 1 and weighted this sample by the posterior probability of cluster membership. Next, we set the value of the dummy variable to 0 and used weight for this sample equal to 1 minus the probability of cluster membership. The covariates were sex, household income (if available), and the normalized birth year (in the case of the UKB cohort). The dependent variable was disease onset. Participants were right censored for a given target disease at their age if the disease was not diagnosed. We calculated separate models for each cross-cohort disease. P-values of the cluster membership variables were adjusted separately for each cohort using the Benjamini–Hochberg method.

In addition, we calculated weighted Kaplan–Meier estimates of MDD-free survival in the various clusters in each cohort. We weighted each participant in each cluster with the corresponding posterior probability of cluster membership.

## Genetic analyses

**Genome-wide association study.** Following site-specific QC measures (see cohort descriptions), Plink 2.0[55] (https://www.cog-genomics.org/plink/2.0/) was used to perform linear regression models to assess the direct effect of each remaining genetic variant on the seven MDD-related clusters that reflected the posterior log-odds of cluster membership. All analyses were adjusted for age, sex, the first ten genetic principal components and site-specific variables (genotyping array in the UKB cohort, geographical region in the THL cohorts). Age was included in the model as a nonlinear variable using cubic splines with knots at ages of 40 and 60 years (R package *splines* v4.1.1, function *bs*). In particular, because of the age span of participants, only the knot at 60 years could be applied in the UKB cohort. Continuous predictor and outcome variables were standardized in the analyses. We employed additive genetic models to assess the contribution of individual genotypes to the dependent variable. We excluded individuals for whom the clustering algorithm lacked sufficient confidence in assigning a predominant cluster, thus concentrating our analysis on those with more clearly defined cluster memberships. More specifically, participants under 60 years of age with a maximum posterior membership probability of no more than 0.25 for any cluster were excluded. Genetic data were available in the UKB ($N = 249{,}167$) and THL ($N = 23{,}786$) cohorts, which were both used to generate the MDD-related clusters. Additionally, genetic data were available in two completely independent cohorts (FinnGen, $N = 277{,}252$ and SHIP, $N = 1126$). We therefore treated the UKB sample as the discovery sample and the latter samples as replication samples. For replication of GWAS loci, a nominal significance level ($p < 0.05$) was assumed.

The FinnGen GWAS analyses were performed with Regenie[56] (version 2.2.4) instead of PLINK 2.0 due to computational constraints in the FinnGen Sandbox pipeline. The parameters for the Regenie analyses were as follows: step 1, bsize 1000; step 2, --bsize 200, --bt false, --apply-rint false, --firth, --approx, --pThresh 0.01, --test additive and --firth-se.

Additional GWAS analysis of the UKB cohort using logistic regression analysis of the binarized presence/absence of disease onset was performed to compare the results of MDD-related multimorbidity clusters to the genetic results of all 86 cross-cohort disease categories used to inform the clusters. All filters and settings were the same as for the cluster membership analysis in the UKB cohort detailed above.

**Post-GWAS analysis.** To assess the impact of SNP results on biological processes, several post-GWAS tools were applied that extract information regarding significant loci, genes, and pathways and report genetic correlations with other phenotypes of interest based on their GWAS summary statistics.

The GWAS summary statistics for all seven MDD-related clusters for each cohort were first processed with FUMA[57] to identify lead SNPs and significant loci. The maximum *p*-value of lead SNPs was set to $5 \times 10^{-8}$, $r^2 \geq 0.6$ was set as the threshold for independent significant SNPs, and the maximum distance between LD blocks of independent significant SNPs was set to 250 kb. Furthermore, MAGMA (v1.10)[58] gene-level analysis was performed to identify putative significant genes using a SNPwise-multi model. We defined the SNP set of each gene including ±10 kb downstream or upstream of the gene, respectively. We used the 1000 Genomes European panel data to evaluate the LD between SNPs. We employed Holm's correction method to adjust the *p*-values of the genes. We assessed the significance of the over-enrichment of MDD-associated genes (identified by Howard et al. [5]) within the genes associated with each cluster by conducting one-sided hypergeometric tests to evaluate whether the association between the cluster genes and MDD genes was stronger than expected by chance. Additionally, we employed Gene Set Enrichment Analysis (GSEA) using the fgsea R package (v1.18)[59] to assess the significance of enrichment of these MDD-associated genes across the clusters. To account for multiple comparisons, we applied Holm's correction method to adjust the *p*-values derived from these tests. In addition, we used the g:Profiler R package (v0.2.3, database version: e110_eg57_p18_4b54a898)[60] for functional enrichment analysis of each cluster's sets of significant genes. We used Gene Ontology (excluding IEA evidence codes) and KEGG biological pathway data sources. We applied the g:SCS method for *p*-value adjustment, and the $p < 0.01$ threshold was used to indicate statistical significance. Using the variety of analysis tools included in the Complex-Traits Genetics Virtual Lab[61] (CTG-VL; https://genoma.io), we additionally assessed the genetic heritability of the clusters, genetic correlations among clusters and genetic correlations with other phenotypes using the LD score regression method (LDSC v1.01)[62]. Genetic correlation is a quantitative statistical parameter reflecting the genetic relationship between two traits. This measure can reflect the pleiotropic action of genes or the correlation between causal loci in two traits, which is especially important for polygenic traits.

**Polygenic risk scores.** A polygenic risk score (PRS) is a genetic measurement that sums an individual's risk-conferring alleles weighted by their estimated effect size for a specific phenotype or disease. The PRS employed in this study was calculated using PRS-CS (v1.0.0), a method that utilizes a high-dimensional Bayesian regression framework and places a continuous shrinkage (CS) prior on SNP effect sizes using GWAS summary statistics and an external linkage disequilibrium (LD) reference panel[63]. Here, the original effect sizes were taken from the UKB GWAS on cluster membership for all seven clusters. The LD reference panel was constructed using a European subsample of the UK Biobank[44]. For the remaining parameters, the default options implemented in PRS-CS were adopted. The PRSs for membership in Clusters 1–7 were calculated in the GWAS samples of the THL and SHIP cohorts. PRSs in the SHIP cohort were correlated with the cluster probabilities, whereas in the THL cohorts, due to the larger sample size, regression analyses between two factors could be performed adjusted for age, sex, batch, region and cohort (Supplementary Data 1).

**Network-based analysis of pleiotropy.** We assessed pleiotropy among the clusters and MDD at the level of functional modules by the following procedure. First, we defined an initial evidence score for the

top genes in each cluster as their negative log-transformed adjusted *p*-value. Next, we applied the Personalized PageRank (also known as Random Walk with Restart, RWR) network propagation algorithm using the interactome network based on STRING (https://string-db.org/; v11.5, filtered to high confidence edges with a combined score cut-off >0.7) to score all protein-coding genes initialized from the top genes of each cluster. Then, we selected the highest-scoring genes in the resulting score rankings by the Kneedle algorithm[64] and identified their nonoverlapping modules using spectral clustering. Next, we similarly initiated network propagation from seed genes significantly associated with MDD based on the gene-level MAGMA analysis results of Howard et al. [5]. Finally, we determined those cluster-specific modules where the propagated MDD scores' sum was statistically significantly higher than according to a null model based on degree-aware permutations of the seed genes in the network. In all RWR experiments, we used a random restart probability of 0.5; however, in accordance with other studies[65], the results were not sensitive to the parameter change.

### Non-genetic risk-factor profiles of MDD-related clusters
Lifestyle and physiological risk factors play a fundamental role in the probability of lifetime MDD. Therefore, it was essential to determine whether one or more of the clusters had a defining risk-factor profile that allowed it to be identified. Apart from basic descriptors such as age, sex, income, and qualifications (Table 1), lifestyle-related factors such as present and past smoking habits and alcohol intake and physiological descriptors such as body mass index, systolic and diastolic blood pressure, pulse rate, C-reactive protein level, and presence of insomnia were investigated. Additionally, neuroticism (as a common personality trait with depression), life stress (as a representation of major negative life events), and current depression score were used as psychological factors to characterize clusters (for availability and descriptive statistics of these descriptors in the individual cohorts, see Supplementary Data 1A).

Linear regression models were constructed for each cluster with Python 3.8 using the statsmodels package (v0.13.1). The corresponding posterior probability log-odds for a given cluster was used as the dependent variable. Two types of regression models were generated: (1) "simple regression models" including one risk factor at a time with age and sex as covariates and (2) "complex regression models" including all available risk factors simultaneously in a single model. Simple regression models allow us to explore the individual effect of each factor, whereas the complex model enables the analysis of joint effects and other multivariate aspects. In the case of complex regression models, a k-nearest neighbour-based imputation method[66] was applied to compute missing values, whereas in the case of simple regression models, only complete samples were used.

### Clustering with a subset of diseases
A natural question that arises is how accurate the clustering will be if not all cross-cohort diseases are available. To assess cluster analysis performance in various limited disease subsets, we recalculated the cluster membership based on various disease subsets of a given size, increasing from only one disease to all cross-cohort diseases: (1) randomly selected diseases; (2) greedily selected, increasingly expanded sets of diseases; (3) a null model based on random cluster membership probabilities; and (4) another null model defined by uniform cluster membership probabilities. We calculated four performance measures, namely, the accuracy and balanced accuracy of the hard clustering (i.e., assigning each individual to the cluster in which its membership probability is highest) and the mean absolute error and mean squared error of the posterior probabilities of cluster membership, averaged over 10,000 random individuals from the UKB cohort selected a hundred times. The greedy variable

selection method was performed as follows. First, we selected a single disease for which the accuracy of cluster analysis (compared to the original clustering of the samples using all cross-cohort diseases) was the highest. Next, we selected the disease from the remaining set of diseases that had the highest accuracy along with the first disease. We began to expand this set, always adding the disease for which the increase in accuracy was the highest. Note that this procedure may result in a suboptimal choice of the best-performing disease subset of a given size. The results for this clustering procedure are provided in Supplementary Fig. S22.

### Reporting summary
Further information on research design is available in the Nature Portfolio Reporting Summary linked to this article.

## Data availability
The following cohorts and biobank data were used for analysis which are available for further research upon application to the data owners: UK Biobank (https://www.ukbiobank.ac.uk/, application number:1602), Catalan Health Surveillance System (CHSS) registry data from all citizens living in the integrated health district of Barcelona-Esquerra ("AISBE") (https://doi.org/10.1186/s12913-019-4174-2), Finnish population surveys (THL, https://thl.fi/en/web/thlfi-en/research-and-development/research-and-projects/previous-research-and-projects), FinnGen project (https://www.finngen.fi/en), and Study of Health in Pomerania (SHIP, https://doi.org/10.1093/ije/dyac034).

## Code availability
The software tools and scripts used in this study are publically available at the following GitHub repository[67]: https://github.com/gezsi/mdd-clustering. While the original data used in our analyses cannot be shared publicly, users can apply these tools to their own datasets to perform MDD-related clustering or execute the full pipeline for clustering individuals with respect to any target disease. For those who have access to the original data, replication of our analyses is possible following the provided instructions in the repository.

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

## Acknowledgements

This research has been conducted using the UK Biobank Resource under Application Number 1602. Linked health data Copyright © 2019, NHS England. Re-used with the permission of the UK Biobank. All rights reserved. This study was supported by the Hungarian National Research, Development, and Innovation Office 2019-2.1.7-ERA-NET-2020-00005 under the frame of ERA PerMed (ERAPERMED2019-108); the Hungarian National Research, Development, and Innovation Office (K 143391, K 139330, PD 146014, and PD 134449 grants); the Hungarian Brain Research Program 3.0 (NAP2022-I-4/2022); and the Ministry of Innovation and Technology of Hungary from the National Research, Development and Innovation Fund, under the TKP2021-EGA funding scheme (TKP2021-EGA-25 and TKP2021-EGA-02). N.E. is supported by the János Bolyai Research Scholarship of the Hungarian Academy of Sciences. Supported by the European Union project RRF-2.3.1-21-2022-00004 within the framework of the Artificial Intelligence National Laboratory. We would like to thank Peter Petschner for his work on identifying a specific issue related to a processing step executed by Plink. SHIP is part of the Community Medicine Research net of the University of Greifswald, Germany, which is funded by the Federal Ministry of Education and Research (grant nos. 01ZZ9603, 01ZZ0103, and 01ZZ0403), the Ministry of Cultural Affairs and the Social Ministry of the Federal State of Mecklenburg-West Pomerania, and the 'Greifswald Approach to Individualized Medicine (GANI_MED)' network funded by the Federal Ministry of Education and Research (grant no. 03IS2061A). Generation of ExomeChip data was supported by the Federal Ministry of Education and Research (grant no. 03Z1CN22). Data collection in SHIP-LEGEND was supported by the German Research Foundation. L.G. was funded by the Deutsche Forschungsgemeinschaft (DFG, German Research Foundation) – grant no. 403694598. This study was supported by the Federal Ministry of Education and Research (BMBF, grant no. 01KU2004) under the frame of ERA PerMed (ERAPERMED2019-108). We thank the participants and investigators of the SHIP study. This study was funded by the Academy of Finland under the frame of ERA PerMed (TRAJECTOME project, ERAPERMED2019-108). We want to acknowledge the participants and investigators of the Finrisk, Health 2000/2011, FinHealth 2017 and FinnGen study. The authors wish to acknowledge CSC – IT Center for Science, Finland, for computational resources. The Catalan cohort was extracted from the Catalan Health Surveillance System database, owned and managed by the Catalan Health Service, with the earnest collaboration of the Digitalization for the Sustainability of the Healthcare (DS3) - IDIBELL group. This study was supported by the Catalan Department of Health (SLD002/19/000002) under the frame of ERA PerMed (ERAPERMED2019-108).

## Author contributions

G.J. and P.A. developed the concept and designed experiments with advice from J.R., I.C., S.V.A., and M.K. G.J., M.K., J.R., I.C., and C.O.S., provided data. A.G., T.N., A.M., and B.B. developed analytical tools. A.G., G.H., N.E., T.N., Z.G., S.V.A., S.B., L.G., K.K., R.G.-C. H.M., T.P., and M.K. collated, cleaned and analysed data. All authors discussed the results and provided critical feedback. A.G., S.V.A., C.O.S., P.A., and G.J. interpreted the results. A.G., S.V.A., X.G., P.A., and G.J. wrote the main paper for which Z.G. gave support. A.G., S.V.A., N.E., and T.N. wrote the Supplementary Information. All authors read and approved the final manuscript.

## Funding

## Competing interests

The authors declare no competing interests.

## Additional information

[1]Department of Artificial Intelligence and Systems Engineering, Budapest University of Technology and Economics, Budapest, Hungary. [2]Department of Psychiatry and Psychotherapy, University Medicine Greifswald, Greifswald, Germany. [3]German Centre for Neurodegenerative Diseases (DZNE), Site Rostock/Greifswald, Greifswald, Germany. [4]Department of Public Health and Welfare, Population Health Unit, Public Health Research Team, Finnish Institute for Health and Welfare, Helsinki, Finland. [5]Department of Pharmacodynamics, Faculty of Pharmaceutical Sciences, Semmelweis University, Budapest, Hungary. [6]NAP3.0-SE Neuropsychopharmacology Research Group, Hungarian Brain Research Program, Semmelweis University, Budapest, Hungary. [7]Clínic Barcelona, Fundació de Recerca Clinic Barcelona - Institut d'Investigacions Biomèdiques August Pi i Sunyer (FRCB-IDIBAPS), Universitat de Barcelona, Barcelona, Spain. [8]Department of Psychiatry and Psychotherapy, Semmelweis University, Budapest, Hungary. [9]Abiomics Europe Ltd., Budapest, Hungary. [10]Institute for Community Medicine, University Medicine Greifswald, Greifswald, Germany. [11]Department of Human Genetics and South Texas Diabetes and Obesity Institute, School of Medicine at University of Texas Rio Grande Valley, Brownsville, TX, USA. [12]Research Program for Clinical and Molecular Metabolism, Faculty of Medicine, University of Helsinki, Helsinki, Finland. [13]These authors contributed equally: Andras Gezsi, Sandra Van der Auwera. [14]These authors jointly supervised this work: Peter Antal, Gabriella Juhasz. ✉e-mail: juhasz.gabriella@semmelweis.hu

