## [Peer Review File · Nature Communications]

Unique genetic and risk-factor profiles in clusters of major depressive disorder-related multimorbidity trajectoriesREVIEWER COMMENTS

Reviewer #1 (Remarks to the Author):

Review of "Unique genetic and risk-factor profiles in multimorbidity clusters of depression-related disease trajectories from a study of 1.2 million subjects".

This is an exciting study presenting results from multiple population cohorts with the aim of identifying clusters of major depressive disorder. The manuscript is very well written and a wide range of analyses have been conducted. The analyses seem to be sound and are described in detail in the main text; detailed descriptions in supplementary methods and data. I would probably need at least days, maybe even weeks to be able to fully grasp all the details that are presented in this article. Since I don't have as much time, I will need to base my evaluation on an overall impression of the work, and based on the details that I have examined, the work seems to be of high quality. This study adds important new information to the literature and the research findings are of interest. In summary, the authors have identified seven clusters that are characterized by differential patterns of temporally assessed disease risk and comorbidity profiles. I have some suggestions for improvement that will hopefully help to further strengthen the paper.

1) I was trying to find a full description of the cohorts in terms of temporal collection of data, clinical diagnosis, and a basic descriptive overview of how comorbidities were determined. I found relevant information in several sections (e.g., main methods, supplementary methods, supplementary tables), but I still think a more broad overview would be helpful. E.g., given that the age range in UKB is limited, how was the onset of MDD diagnosis or depressive symptoms assessed? How many MDD cases in each dataset? What is the temporal nature of the data in each of the datasets? Is it derived from retrospective reports, electronic health record data, or multiple assessments as part of the study design? Figure 1 is helpful but could be extended by including information on cohort design and description of data, even if it this can only be provided in the supplementary.

2) From what I understand, genetic and environmental risk factors were included in the machine learning analyses that were used to derive the clusters. How was genetic and environmental risk operationalized? And would it be double dipping to then conduct a GWAS of the derived clusters?

3) Regarding the GWAS of the derived clusters, from what I understand, the clusters are modelled as the dependent variable and the genotype as the independent variable. I could not easily find more information on this in the supplementary material. What was the reference group of the seven clusters? How exactly was genotype modelled? As a binary variable or were the 3 genotype groups used as the outcome?

4) To me, the term "modifiable risk factors" implies a causal relationship between these variables and the outcome (if we modify the risk factor, the risk would be reduced). I think that is too strong, i.e., no causal analyses were conducted. Also, many modifiable risk factors, such as smoking, drinking or SES, are heritable and may not in fact be modifiable. I would prefer different terminology, e.g., epidemiological risk factors, but will leave this to the authors to decide

5) From the supplementary tables, I could see that only a limited number of items was used to identify current depressive symptoms in UKB. A better option would be to use the PHQ-9, see <https://pubmed.ncbi.nlm.nih.gov/31530331/> for a description. Was CIDI diagnosis also used as a variable?

6) In the supplementary table, the sample size of UKB is described as UKB GWAS (n=247,325) and UKB ALL*(n=364,008). These numbers do not match the number provided in the main text. "To identify MDD-related clusters based on disease trajectories, 502,504 participants who had available disease onset information for 1,127 ICD-10 categories were included (for details, see the following section: Cross-cohort disease categories and relevance scores)." Please explain.

Reviewer #2 (Remarks to the Author):

The Gezsi et al. study leverages several large and some minor biobanks to identify and characterize MDD multimorbidity trajectories. The ambition is to gain insight into clinical, etiological as well as (potentially modifiable) environmental risk factors of MDD. The strategy is to

use both somatic and possibly other mental disorders co-occurring with MDD to identify and characterize biological subtypes of MDD. Gezsi et al. first selects disorders that frequently co-occur with MDD, and next assigns a scores of the MDD-related multi-morbidities to each participant, which in turn is used to identify or define seven clusters. The corresponding MDD-related disease-trajectories were contrasted with respect to disease prevalence and age-at-diagnoses, and GWAS analyses were performed to identify and compare genomic markers across the clusters.

Longitudinal, analytical approaches considering life-time disease-occurrence as performed in this study are obvious supplements or alternatives to the clinical cross-sectional evaluation in the quest to identify subtypes, causes or pathological processes of disease. While the manuscript by Gezsi et al. describes a novel and interesting approach in the realm of MDD, the study has several caveats that dampens my enthusiasm somewhat.

It remains unclear how precisely the 86 'MDD-related disorders' are identified and selected (simply stated in line 121); reference to Online Methods (Fig. 1) does not inform either. The selection is key to the interpretation of later findings and should be pinned out in detail. The 86 'MDD-related disorders' might be the most prevalent conditions in the cohorts, and only subsequently linked to or analyzed in the context of MDD. Alternatively, the 86 disorders might be the most prevalent disorders among MDD patients, and therefore used for trajectory building. In either case, it is relevant (to say the least) to examine the relation between patterns of prevalence disease in the cohorts and the MDD individuals in the cohorts. The repeated use and the importance ascribed to the terms 'MDD-related' indicate the latter of the two options, while the text generally hints at the former option and consistently refers to the full cohorts; in fact, the number of participants in the cohorts are mentioned several times while those of MDD patients are not – curiously, one has to consult tables to know the numbers of MDD patients going into the analyses.

Whether or not the selected disorders are chosen b/c of enrichment among MDD patients relative to non-MDD participants, is key to the study rationale. Using the prevalence in the cohorts rather than among MDD-participants would bias away from MDD-etiology and -stratification, and simply provide insight into the general disease-constellations in the population on which MDD occurs. Contrary, a strategy focusing on MDD-related disorders (consistent with the intention stated by the authors) would enrich for genetic or environmental factors contributing specifically for MDD or MDD-trajectories. In this case, it would be relevant to include MDD-enrichment into the subsequent analyses.

A non-neglectable subset of MDD patients go on to develop bipolar or schizophrenia, and many have been diagnosed with other psychiatric disorders before the MDD diagnosis. Given this extensive co-occurrence of psychiatric disorders, it would have been relevant to include a separate analysis of MDD in psychiatric trajectories, and to examine their relation to the overall / general trajectories.

The genetic analyses are hard to interpret without a more detailed understanding of the selection criteria of the trajectory disorders. It remains unclear whether the GWAS findings are driven by differences in the patterns of comorbidity-prevalence between clusters or (as the study rationale aims at) by the hypothesized MDD-subtype. Several large cohorts are included in the analyses and thus would allow authors to use the results of the genetic clusters/trajectories analyses in one cohort to predict cluster membership of individuals in another cohort, and to contrast their predicted and actual comorbidity patterns.

While the authors are aware of the differences between cohorts such as age-distribution, observation periods, and healthcare provision, it is unclear how the possible effects are handled. In fact, participants seem to be aligned by age (age=0 years; Fig. 1B) rather than by birth-year. This implies that MDD-individuals in the same age-bin (e.g. age 0-20 yr) may have been born and diagnosed years apart and even with different diagnostic classifications, clinical practice and healthcare provision. Smoking habits (strongly linked to MDD-liability) have changed considerably over the past decades and therefore also somatic consequences of smoking have changed.

The genetic analyses contrasting two clusters will by necessity be conditioned by different prevalence of early-vs-late onset disorders. More importantly, it is unclear whether the genetic

analyses of all cluster-individuals inform on MDD subtypes or simply reflect differences in prevalence of other disorders in the clusters. Authors could have considered to simply estimate the SNP heritability of MDD individuals (only) across clusters to quantify a possible genetic component.

Reviewer #3 (Remarks to the Author):

General

The authors set out to identify subtypes of MDD using information from multimorbidities in patients with MDD, identifiable in large biobanks such as UKBiobank and FinnGen. They first use Bayesian networks to identify a minimal, nonredundant set of MDD-related multimorbidity trajectories from the entire medical histories, then assess patient similarities using this filtered set of trajectories, and finally cluster patients across cohorts using a privacy preserving Bayesian approach. The paper is well-written with excellent figures illustrating the concepts and results. My experience is with MDD genetics, and I am not familiar with some of the Bayesian approaches the authors have made use of, therefore my comments would be mostly on the genetics side, with some questions on the methods.

Comments

1. Identifying strongly relevant multimorbidities: I am unfamiliar with the inhomogeneous dynamic Bayesian network approach, but after reading the explanation in Methods, I am still unclear as to how the authors could determine the direction of arrows (direct edge from X to Y using the authors' words) pointing from diseases to each other in the system, without testing for causality (which is itself really difficult) or interactions (another difficult thing). I didn't see any tests for either, am I missing something here?
2. weighted directed MDD-related multimorbidity scores: this seems to be a sum score of each individual's propensity to get MDD given their other diseases at each time t, weighted (0,1) by whether they actually had a first diagnosis of each other disease at each time t. This seems a completely new metric to me, and I would like to see more characterisation of this metric. Why is this the right metric to use? How is onset of MDD being taken into account? How does age range in the different cohorts and availability of data on disease onset affect this score? For example, do the individuals in UKB (age range 37-93) have disease records going back to when they are much younger, for identifying earlier onset diseases (seems not from Fig 2, as most disease onsets are after 30)? This is also supported by Fig S3, which shows that the onsets of diseases in the CHSS cohort (the youngest) seem earlier? As strong-relevance and this score form the basis for clustering and all further analysis, I find it hard to assess the merits of further analyses without understanding these concepts. All my comments from this point on are assuming these two metrics are well-characterised, which I think the authors ought to demonstrate.
3. GWAS on clusters: something is going on in chromosome 6 across GWAS on all clusters - the fact the entire chromosome is lighting up seems an impossible real finding to me. The authors wrote about immune profiles in the main text, but does not address whether this signal is likely an artefact. In Fig S15, the authors showed the GWAS performed in FinnGen and I don't see the entire chromosome 6 lighting up. Table 3 further showed that results from FinnGen mostly do not replicate findings from the UKB analysis (<50% for loci and <15% at gene level in all clusters, though the authors wrote successful replication in the main text). r_G between the UKB clusters and FinnGen clusters are also moderate at best (I would like to see heritability estimates also), and the authors didn't give values of variance explained by PRS from UKB on the smaller cohorts (only the p values, from which I can't tell how good the predictions are). Also note the replicated genes are also mostly in the MHC region, which is highly influenced by population structure and therefore potentially artefacts.

Overall

The authors showed a high level of sophistication in their work that I feel is not matched by rigour to characterise new metrics they are proposing and findings that seem to harbour artefacts. I am not currently convinced by their methods and findings, but I am happy to be proven wrong if they are able to demonstrate that the metrics are well-suited for disease trajectory characterisation, and their clusters aren't just tagging population structure.

Unique genetic and risk-factor profiles in multimorbidity clusters of depression-related disease trajectories from a study of 1.2 million subjects

NCOMMS-23-36313-T

REVIEWER COMMENTS

Reviewer #1. Quantitative genomics and depression.

Reviewer #1 (Remarks to the Author):

Dear Reviewer #1,

We sincerely appreciate your time and effort in reviewing our manuscript, "Unique genetic and risk-factor profiles in multimorbidity clusters of depression-related disease trajectories from a study of 1.2 million subjects." We are grateful for your insightful comments and constructive feedback, and we anticipate that they will significantly enhance the strength and impact of our paper.

Review of "Unique genetic and risk-factor profiles in multimorbidity clusters of depression-related disease trajectories from a study of 1.2 million subjects".

This is an exciting study presenting results from multiple population cohorts with the aim of identifying clusters of major depressive disorder. The manuscript is very well written and a wide range of analyses have been conducted. The analyses seem to be sound and are described in detail in the main text; detailed descriptions in supplementary methods and data. I would probably need at least days, maybe even weeks to be able to fully grasp all the details that are presented in this article. Since I don't have as much time, I will need to base my evaluation on an overall impression of the work, and based on the details that I have examined, the work seems to be of high quality. This study adds important new information to the literature and the research findings are of interest. In summary, the authors have identified seven clusters that are characterized by differential patterns of temporally assessed disease risk and comorbidity profiles. I have some suggestions for improvement that will hopefully help to further strengthen the paper.

1) I was trying to find a full description of the cohorts in terms of temporal collection of data, clinical diagnosis, and a basic descriptive overview of how comorbidities were determined. I found relevant information in several sections (e.g., main methods, supplementary methods, supplementary tables), but I still think a more broad overview would be helpful. E.g., given that the age range in UKB is limited, how was the onset of MDD diagnosis or depressive symptoms assessed? How many MDD cases in each dataset? What is the temporal nature of the data in each of the datasets? Is it derived from retrospective reports, electronic health record data, or multiple assessments as part of the study design? Figure 1 is helpful but could be extended by including

information on cohort design and description of data, even if it this can only be provided in the supplementary.

Response to Question 1:

Thank you for pointing out this important aspect. A comprehensive description of the incorporated cohorts is indeed of essential importance for the proper interpretation of the methods, the resulting clusters, and the downstream outcomes. To enhance both clarity and transparency in our methodology, we collected the requested essential information for the participating cohorts utilized in the initial clustering process. This offers a more broad overview of the cohort structures, as detailed in Supplementary Table S1B. Moreover, we also included two figures illustrating the distribution of assigned diagnoses per year for all conditions and specifically for MDD only within the UK Biobank dataset (see Supplementary Figure S1A&B). As expected, the data density improves notably after the 1990s, aligning with the trend of increased digitalization. Nevertheless, chronic diseases diagnosed at an early age remain within the system and are captured by our data. Similarly, our dataset comprehensively covers diseases that predominantly manifest in older age, ensuring their inclusion in our analyses.

Figure S1. The distribution of assigned diagnoses per year within the UK Biobank dataset for all conditions (A), and specifically for MDD (B).

Additionally, we determined the cumulative count of disease diagnoses with increasing age across all three discovery cohorts and observed highly similar trends (see Supplementary Figure S2). This

similarity suggests that the cohorts effectively capture comparable age-related multimorbidity patterns.

Figure S2. The cumulative count of disease diagnoses with increasing age across the three discovery cohorts (UKB, CHSS, THL).

2) From what I understand, genetic and environmental risk factors were included in the machine learning analyses that were used to derive the clusters. How was genetic and environmental risk operationalized? And would it be double dipping to then conduct a GWAS of the derived clusters?

Response to Question 2:

To clarify, genetic and environmental risk factors were deliberately not included in the initial clustering analyses to derive the clusters. Instead, we focused exclusively on clinical data, specifically retrospective disease onset information, to derive these clusters. This approach was taken as we aimed to demonstrate that these clusters possess distinct genetic backgrounds and non-genetic risk factor profiles, independent of their initial identification.

Although including genetic and environmental factors in the clustering process could be a rational strategy, considering they are root causes of multimorbidity, we opted for a different approach. Our primary goal was to derive the clusters by focusing solely on clinical data (disease onset information), and establish that the resulting clusters have unique genetic profiles and are transferable across different populations. This is evident from our analysis where, based on only clinical data, we computed clusters in other populations (specifically FinnGen and SHIP cohorts) and found considerable overlap between significant genetic variants among these populations. This suggests

that the cluster definitions, derived from clinical data alone, have robust and definable genetic underpinnings.

For further details on our methodology, please refer to the sections of our paper where we discuss the selection of diseases for cluster analysis based on their strong relevance to MDD (sections "Assessing diseases strongly relevant to MDD" (Lines 640-693) and "Cross-cohort disease categories and relevance scores" (Lines 694-720)), and where we describe the clustering of the participants based on the weighted direct MDD-related multimorbidity scores derived from disease onset information (section "Clustering of participants" (Lines 721-765)).

Regarding your question on 'double dipping' in conducting a GWAS of the derived clusters, since the initial clustering was not based on genetic data, conducting a GWAS on these clusters does not constitute double dipping. It rather serves as an independent validation and exploration of the genetic underpinnings of the clinically derived clusters.

3) Regarding the GWAS of the derived clusters, from what I understand, the clusters are modelled as the dependent variable and the genotype as the independent variable. I could not easily find more information on this in the supplementary material. What was the reference group of the seven clusters? How exactly was genotype modelled? As a binary variable or were the 3 genotype groups used as the outcome?

Response to Question 3:

In our analysis, the posterior log-odds of the cluster memberships were used as the target (dependent) variables for both the GWAS and the non-genetic risk-factor profiling. This approach allowed us to circumvent the need for defining a specific reference group or controls, as all participants were included in the GWAS for each cluster. Currently, this information is mentioned only in the captions of Table 2 and Figure 4 and in the Online methods. As it is crucial, we acknowledge that its omission from the main text may have led to some ambiguity. Therefore, we include this information in the revised manuscript in the "GWAS analysis of MDD-related multimorbidity clusters in the UKB cohort identifies immune system-related genetic profiles" section (Lines 233-235):

"To explore the genetic background of the clusters, we conducted GWAS analyses in the UKB cohort (N=249,167), where the posterior log-odds of the cluster memberships were used as the target variables."

Regarding the modeling of genotype, we employed an additive genetic model. We include a specific mention of the use of an additive genetic model in the revised manuscript (Lines 800-801):

“We employed additive genetic models to assess the contribution of individual genotypes to the dependent variable.”

For more detailed information on our GWAS methodology, please refer to the relevant sections of the Online methods, where these aspects are discussed in detail (“Genetic analyses - Genome-wide association study” (Lines 790-819)).

4) To me, the term “modifiable risk factors” implies a causal relationship between these variables and the outcome (if we modify the risk factor, the risk would be reduced). I think that is too strong, i.e., no causal analyses were conducted. Also, many modifiable risk factors, such as smoking, drinking or SES, are heritable and may not in fact be modifiable. I would prefer different terminology, e.g., epidemiological risk factors, but will leave this to the authors to decide

Response to Question 4:

Thank you for your insightful observation regarding our use of the term “modifiable risk factors.” We acknowledge your concerns, as you rightly pointed out, no causal analyses were conducted, and some of these factors, like smoking, drinking, or socioeconomic status, have heritable components and may not be entirely modifiable.

*To avoid any ambiguity and to more accurately reflect the nature of these factors, we agree with your suggestion. We will replace the term “modifiable risk factors” with “**non-genetic risk factors**” throughout our paper. This terminology will effectively distinguish these factors from genetic ones and align more closely with the scope and methodology of our research.*

5) From the supplementary tables, I could see that only a limited number of items was used to identify current depressive symptoms in UKB. A better option would be to use the PHQ-9, see <https://pubmed.ncbi.nlm.nih.gov/31530331/> for a description. Was CIDI diagnosis also used as a variable?

Response to Question 5:

Thank you for the useful suggestion. We had not included these depression phenotypes as the main depression descriptors for the clusters in the paper, because their sample size is considerably smaller than that of the 4-item current depression score (Supplementary Table S1A). This is because between July 2016 and July 2017 only a subset of 157,366 participants completed the online follow-up mental health questionnaire (MHQ), which included questions regarding issues such as depression, anxiety, traumatic events or self-harm behaviours (Davis et al., 2020). In particular, the PHQ-9 score is accessible for a sample size of $n = 157,321$ within the entire UK Biobank dataset. To ensure consistency with our prior analyses, we refined the dataset to include only those individuals who were also

involved in the genetic analysis. Within this refined subset, there were 348 cases and 78,453 controls for the current depressive episode based on the CIDI criteria, and 11,349 cases and 66,161 controls for the lifetime depression phenotype based on the CIDI criteria (for definitions see Supplementary Methods). For some participants, the CIDI-based depression-status was undeterminable. This is in contrast to the EHR-based ICD-10 MDD diagnosis, where 220,315 controls and 28,851 cases are present in this sub-sample.

Nevertheless, following your question, we decided to run a series of analyses in the UKB dataset to see if using PHQ-9 and CIDI phenotypes yields the same or similar results as using the current depression score and EHR-based ICD-10 MDD diagnosis, as we did in our study. We have run linear regression models in UKB for cluster membership probability log-odds as the outcome variables, with each distinct depression phenotype definitions separately as a predictor variable. Methods and results are now detailed in lines 213-273 of Supplementary Methods. In summary, cluster membership showed a consistent association pattern with depression with minor differences according to the types of definitions. Namely, Cluster 1-3 showed a robust negative, while Cluster 5-6 a strong positive association with the different depression phenotypes. Cluster 4 showed no major association with them, and Cluster 7 showed negative association only with the different current depression descriptors. By these analyses we demonstrated good agreement between CIDI/PHQ-9 and the depression phenotypes we used, suggesting that our results regarding cluster membership are generalisable using different depression definitions.

Figure SM8 A-E. Regression coefficient of depression in linear regression models for cluster membership probability log-odds as the target variable. Predictors in each model were sex and age, in addition to the respective depression phenotype. Standardized regression coefficients of each depression phenotype, with a 95% confidence interval, are displayed for each cluster membership.

Reference:

Davis KAS, Coleman JRI, Adams M, et al. (2020) Mental health in UK Biobank – development, implementation and results from an online questionnaire completed by 157 366 participants: a reanalysis. *BJPsych Open* 6(e18): 1-8. <https://doi.org/10.1192/bjo.2019.100>

6) In the supplementary table, the sample size of UKB is described as UKB GWAS (n=247,325) and UKB ALL*(n=364,008). These numbers do not match the number provided in the main text.

“To identify MDD-related clusters based on disease trajectories, 502,504 participants who had available disease onset information for 1,127 ICD-10 categories were included (for details, see the following section: Cross-cohort disease categories and relevance scores).” Please explain.

Response to Question 6:

Thank you for pointing out the discrepancy in the sample size numbers between the main text and the supplementary table. To clarify:

- 1. Initial participant count: In the UK Biobank, we initially included 502,504 participants who had available disease onset information. This was the broader set of participants used for the identification of MDD-related clusters based on disease trajectories.*
- 2. Subset by filtering out patients with uncertain MDD-related disease trajectories: To refine our analysis, we sought to ensure the robustness of cluster assignments by excluding participants for whom the clustering algorithm demonstrated low confidence across all clusters. Specifically, we excluded participants who were both under 60 years of age and whose maximum posterior membership probability did not exceed 0.25 for any cluster. This threshold was chosen to remove individuals for whom the algorithm could not confidently assign a predominant cluster, thereby focusing our analysis on participants with more definitive cluster memberships. This exclusion criterion resulted in a subset of $N = 364,008$ participants. This subset was used for comparing clusters and deriving age-specific differences. This subset was also the base set for genetic analysis.*
- 3. GWAS Quality Control: In further refining our analysis for the GWAS, we conducted a comprehensive quality control process. This included kinship filtering to select a maximal set of unrelated individuals, sex check and heterozygosity outlier detection, the inclusion of participants based on up-to-date consent, and filtering participants with missing data on critical variables such as sex, age, or genotyping array. These steps resulted in a final cohort of $N = 249,167$ participants for the genetic analysis, drawn from the broader subset of $N = 364,008$ individuals.*

We recognize the need for clarity in presenting these figures and provide a more explicit explanation of these participant subsets in the revised version of the manuscript (see Lines 751-759 and 800-805).

Reviewer #2. Psychiatry, risk factors and genetics.

Reviewer #2 (Remarks to the Author):

Dear Reviewer #2,

We sincerely appreciate your time and effort in reviewing our manuscript, "Unique genetic and risk-factor profiles in multimorbidity clusters of depression-related disease trajectories from a study of 1.2 million subjects." We are grateful for your insightful comments and constructive feedback, and we anticipate that they will significantly enhance the strength and impact of our paper.

The Gezi et al. study leverages several large and some minor biobanks to identify and characterize MDD multimorbidity trajectories. The ambition is to gain insight into clinical, etiological as well as (potentially modifiable) environmental risk factors of MDD. The strategy is to use both somatic and possibly other mental disorders co-occurring with MDD to identify and characterize biological subtypes of MDD. Gezi et al. first selects disorders that frequently co-occur with MDD, and next assigns a scores of the MDD-related multi-morbidities to each participant, which in turn is used to identify or define seven clusters. The corresponding MDD-related disease-trajectories were contrasted with respect to disease prevalence and age-at-diagnoses, and GWAS analyses were performed to identify and compare genomic markers across the clusters.

Longitudinal, analytical approaches considering life-time disease-occurrence as performed in this study are obvious supplements or alternatives to the clinical cross-sectional evaluation in the quest to identify subtypes, causes or pathological processes of disease.

We appreciate your insightful observation on the analytical approach of our study. In response to your comment, we would like to clarify a key aspect of our methodology.

It's crucial to clarify that our method does not "select[s] disorders that frequently co-occur with MDD". Instead, it selects a minimal set of disorders, the collective status of which encapsulates the entirety of disease-related information about MDD risk. For a more comprehensive explanation, please refer to our detailed response to the next comment regarding the selection of MDD-related disorders.

Our study, while it utilizes disease onset information, is neither strictly longitudinal nor cross-sectional in nature. The disease onset data were collected retrospectively from various sources, including electronic health records, UK Biobank baseline assessments, registers of primary health care and health care inpatient visits, and registers of specialist outpatient visits. This retrospective collection allows us to construct a temporal map of disease occurrences and relationships without the constraints of a traditional prospective study.

In addition to the approach outlined for disease onset information, it is important to clarify the treatment of non-genetic risk factors in our study. These factors, earlier referred to as modifiable risk factors, were collected in a cross-sectional manner. Our objective here was to conduct an association analysis to determine the non-genetic risk factor profile specific to each of the identified clusters. This analysis essentially provided a snapshot of all participants assigned to a particular cluster at a specific point in time, which corresponds to the period when the risk factors were assessed. This approach allowed us to identify and understand the distinct non-genetic characteristics (including many risk factors of non-communicable diseases) that are associated with each cluster, enhancing our understanding of the overall multimorbidity profiles within the context of Major Depressive Disorder.

Your comment highlights an essential distinction in the methodological approach of our study, and we make this clearer in the revised manuscript to avoid any confusion regarding the nature of our data and analyses. For that, we explicitly mention in the revised text that non-genetic risk factors were collected cross-sectionally (Lines 326-329, the inserted text is underlined):

“To assess the non-genetic risk factors collected cross-sectionally within the clusters, we examined associations of behavioural and physiological factors with the MDD-related clusters. This analysis determined the specific risk factor profiles for each identified MDD-related cluster, offering a snapshot of all participants at the time when these factors were evaluated.”

While the manuscript by Gezi et al. describes a novel and interesting approach in the realm of MDD, the study has several caveats that dampens my enthusiasm somewhat.

It remains unclear how precisely the 86 ‘MDD-related disorders’ are identified and selected (simply stated in line 121); reference to Online Methods (Fig. 1) does not inform either. The selection is key to the interpretation of later findings and should be pinned out in detail. The 86 ‘MDD-related disorders’ might be the most prevalent conditions in the cohorts, and only subsequently linked to or analyzed in the context of MDD. Alternatively, the 86 disorders might be the most prevalent disorders among MDD patients, and therefore used for trajectory building. In either case, it is relevant (to say the least) to examine the relation between patterns of prevalence disease in the cohorts and the MDD individuals in the cohorts.

The repeated use and the importance ascribed to the terms ‘MDD-related’ indicate the latter of the two options, while the text generally hints at the former option and consistently refers to the full cohorts; in fact, the number of participants in the cohorts are mentioned several times while those of MDD patients are not – curiously, one has to consult tables to know the numbers of MDD patients going into the analyses.

We appreciate your critical assessment of our study's selection process for the 86 ‘MDD-related disorders’. You are correct in noting that this selection is crucial for interpreting our findings. We covered these aspects in the Results (see section “Dynamic Bayesian network analysis reveals seven

MDD-related multimorbidity clusters” (Lines 121-137)) and in the Online methods (see section “Assessing diseases strongly relevant to MDD” for the methodological background (Lines 641-693) and section “Cross-cohort disease categories and relevance scores” for the description of disease selection (Lines 695-720)). We summarized these paragraphs below and also extended them in the relevant sections based on your question.

Our aim was to identify a minimal, nonredundant set of MDD-related multimorbidity trajectories that conveys all relevant information about MDD in a subject’s entire medical history from three large population cohorts, including both MDD and non-MDD cases. Based on our previous research (Marx et al. 2017), we hypothesized that this information facilitates the identification of biologically and clinically informative depression subtypes as well as their distinct neurobiological and genetic backgrounds (for details see our response below regarding MDD-enrichment analysis).

Therefore, we need to clarify that the selection of these 86 diseases was not determined by the presence of MDD. Instead, our approach was to identify multimorbidities directly related to the susceptibility of MDD in the entire population, independent of an MDD diagnosis. This means that during the selection of these 86 diseases, their prevalence was not explicitly considered in either the overall cohort or the subpopulation of MDD patients. For instance, Type 2 diabetes mellitus (ICD-10: E11) has a relatively high prevalence in the general population (UKB: 6.6%, CHSS: 7.1%, THL: 13.5%) and an even significantly higher prevalence within the subpopulation with depression (UKB: 10.3%, CHSS: 16.8%, THL: 17.3%). However, given its lack of a direct relationship to MDD, as indicated by a mean probability of strong relevance ranging from 0.0 to 0.2 across various cohort and intervals, this disease was not included among those directly related to MDD.

Nevertheless, in the preliminary phase of our study, we conducted an initial filtering of diseases, selecting only those with a prevalence exceeding 1% in either the entire cohort or the subset of participants diagnosed with depression. This process was independently applied to each cohort (UKB, THL, CHSS). The primary objective of this pre-filtering step was to exclude rare disorders, as our goal was to identify general multimorbidity trajectory clusters that are broadly applicable. Consequently, this initial filtering led to differing numbers of diseases being considered across each cohort, namely 266 disorders for UKB, 356 for CHSS, and 339 for THL. We make this clear in the revised manuscript to avoid any confusion (see Lines 698-702).

After this pre-filtering step, we employed dynamic Bayesian networks within a Bayesian statistical framework on the remaining disease variables to identify a minimal, nonredundant set of MDD-related multimorbidities, effectively encapsulating all relevant information about MDD throughout a subject’s complete medical history. This filtered 86 disorders includes all multimorbidities with nonmediated causal relationships to MDD and those with potential shared genetic and modifiable factors. Diseases were selected based on having a posterior probability of strong relevance with respect to MDD greater than 0.5 in at least one time interval for at least one cohort, ensuring that

each disease is consistently represented across all cohort datasets. These criteria ensure that only diseases with a significant probability of being strongly related to MDD are included while also guaranteeing their universal presence across the diverse cohort datasets, providing a robust foundation for comparative analysis. We clarify this in the revised manuscript (see Lines 707-710). Furthermore, we have expanded our manuscript to include a more comprehensive explanation of the concept of strong relevance and related terms, specifically within the framework of Bayesian analysis (see Lines 669-680). Also, in the revised version of our paper, we have provided a detailed characterization of the weighted direct MDD-related multimorbidity burden score utilized for clustering. This includes a detailed exploration of how the score associates with both the prevalence and onset of MDD, as outlined in the "Characterization of the weighted directed MDD-related multimorbidity score" section found in the Supplementary Methods.

The clusters were computed for the entire population, each corresponding to distinct trajectories of MDD-related multimorbidity. This methodology was deliberately chosen to enable the assignment of participants to clusters irrespective of an MDD diagnosis. Our main aim was to derive a broadly applicable, general clustering framework where each cluster represents a unique multimorbidity trajectory related to MDD.

Reference:

Marx et al. (2017) Comorbidities in the diseasome are more apparent than real: What Bayesian filtering reveals about the comorbidities of depression. PLoS Comput Biol 13(6): e1005487

Whether or not the selected disorders are chosen b/c of enrichment among MDD patients relative to non-MDD participants, is key to the study rational. Using the prevalence in the cohorts rather than among MDD-participants would bias away from MDD-etiology and -stratification, and simply provide insight into the general disease-constellations in the population on which MDD occurs. Contrary, a strategy focusing on MDD-related disorders (consistent with the intention stated by the authors) would enrich for genetic or environmental factors contributing specifically for MDD or MDD-trajectories. In this case, it would be relevant to include MDD-enrichment into the subsequent analyses.

In response to your comment, we state that our study comprehensively incorporated MDD enrichment into our analyses in four key ways: firstly, by examining the incidence and onset age distribution of MDD cases across the clusters; secondly, by evaluating the extent of pleiotropy between MDD and the identified clusters; thirdly, by conducting a case-only analysis on the UKB cohort, focusing exclusively on individuals diagnosed with MDD and performing a GWAS on these filtered clusters; and fourthly, by evaluating depression related phenotypes (current depression score, PHQ-9, and CIDI).

Specifically, the next to last paragraph of "Dynamic Bayesian network analysis reveals seven MDD-related multimorbidity clusters" section (Lines 196-199) and Figure 3 B&C in our manuscript demonstrate how the occurrence of MDD in the clusters was analyzed in our study. This analysis showed a marked difference between the incidence rate and onset age distribution of MDD across the clusters.

Besides, the enrichment of genetic or environmental factors specifically contributing to MDD was indeed one of our primary aims, which we sought to achieve through a comprehensive, systems-based analysis. As stated in our manuscript, "The fundamental rationale of our method was that defining the clusters based only on the strongly relevant diseases to MDD focuses the clusters' profile on pleiotropic genetic and environmental factors of these diseases. In contrast, factors affecting MDD only through other diseases are diminished." The Bayesian Network-based approach allowed us to concentrate on diseases directly relevant to MDD. Earlier, we showed that the direct multimorbid neighborhood of a disorder (specifically depression) corresponds to direct relationships in the molecular interactome (Marx et al. 2017). In our current study, we also assessed the extent of pleiotropy among the clusters and MDD, and found several cluster-specific genes and functional modules of the human interactome that were pleiotropic with MDD (see the final paragraph of "GWAS of MDD-related multimorbidity clusters in the UKB cohort identify immune-system related genetic profiles" section (Lines 310-324)).

Moreover, in response to your suggestion, we have now conducted a case-only analysis on the UKB cohort, focusing exclusively on individuals diagnosed with MDD and conducting a GWAS on these filtered clusters in a manner similar to our approach with the full population (for details see our response below regarding MDD case-only analysis).

Finally, we extended our analysis to include various depression phenotypes, such as current depression score, current PHQ-9 score, current and lifetime CIDI, and clinically diagnosed MDD (for details see our response below regarding MDD case-only analysis).

Reference:

Marx et al. (2017) Comorbidities in the diseasome are more apparent than real: What Bayesian filtering reveals about the comorbidities of depression. PLoS Comput Biol 13(6): e1005487

A non-neglectable subset of MDD patients go on to develop bipolar or schizophrenia, and many have been diagnosed with other psychiatric disorders before the MDD diagnosis. Given this extensive co-occurrence of psychiatric disorders, it would have been relevant to include a separate analysis of MDD in psychiatric trajectories, and to examine their relation to the overall / general trajectories.

Thank you for your suggestion regarding the analysis of MDD in the context of psychiatric diseases. To address this, we conducted a dedicated analysis focusing specifically on psychiatric diseases (Chapter V. of ICD-10: F00-F99) to assess their temporal disease patterns across the identified clusters. This analysis revealed significant differences among the clusters in terms of the onset and prevalence of various psychiatric disorders (see figure below).

Notably, we observed that many psychiatric disorders had distinctive onset patterns in different clusters. For example, the onset age of many disorders such as alcohol-related disorders, phobic and other anxiety disorders, bipolar disorder, and nicotine dependence was reduced in Cluster 5, and schizophrenia showed a substantially lower onset age in Clusters 5 and 7. This indicates a marked variation in the psychiatric disease trajectories among the MDD-related clusters, underscoring the heterogeneity within MDD-related multimorbidity.

Figure S6. Temporal disease patterns of the complete set of psychiatric diseases in the clusters according to UKB (N = 502,504). The average onset ages of the psychiatric diseases (Chapter V. of ICD-10: F00-F99) per line according

to the seven clusters in the UKB cohort. The node color indicates the clusters and the node size is proportional to the observed prevalence of the disease in the cluster.

We include this figure in the revised version of the supplementary material and indicate the results of this specific analysis in the main text (Lines 199-202):

“Focusing on the complete set of psychiatric diseases (Chapter V. of ICD-10: F00-F99) to assess their temporal disease patterns across the identified clusters, analysis revealed significant differences among the clusters in terms of the onset and prevalence of various psychiatric disorders (see Supplementary Fig. S6).”

The genetic analyses are hard to interpret without a more detailed understanding of the selection criteria of the trajectory disorders. It remains unclear whether the GWAS findings are driven by differences in the patterns of comorbidity-prevalence between clusters or (as the study rationale aims at) by the hypothesized MDD-subtype. Several large cohorts are included in the analyses and thus would allow authors to use the results of the genetic clusters/trajectories analyses in one cohort to predict cluster membership of individuals in another cohort, and to contrast their predicted and actual comorbidity patterns.

We appreciate your highlighting the potential to use genetic clusters and trajectory analyses from one cohort to predict cluster membership in another, and to compare predicted versus actual comorbidity patterns. This indeed represents an interesting open question and a promising direction for future research.

However, in the current study, we opted to concentrate on a different but complementary goal: our primary objective was to elucidate the genetic underpinnings of the clusters identified based on clinical data, where these clusters reflect different temporal trajectories of the MDD-related multimorbidity burden throughout the lifespan. In doing so, we aimed not just to identify genetic variants associated with these clusters, but more importantly, to infer the underlying biological mechanisms in order to gain insights into the genetic and biological basis of the observed multimorbidity patterns. We did not aim to use genetic data as a predictor for cluster membership, as such an approach would have been overambitious given the relatively low heritability of the cluster memberships and in general, the scope of our study. Moreover, by focusing on clinical data (disease onset information) that is more readily available and employing relatively stable ICD-10 level-3 codes, we aimed to mitigate the impact of the variations between cohorts.

Moreover, a key goal of our research was to demonstrate the transferability of these clusters across different populations. To this end, we performed an independent GWAS analysis in the FinnGen cohort. This allowed us to examine whether the genetic background of the clusters identified in our

primary cohort held true in an entirely different population (see “Validation of MDD-related multimorbidity profiles at the genetic and non-genetic risk-factor levels” section (Lines 367-382) and Table 3). Additionally, as a further validation step, we conducted polygenic risk score (PRS) analyses in two separate cohorts, namely the Study of Health in Pomerania (SHIP) (Lines 405-410) and the Finnish Institute for Health and Welfare (THL) cohorts (Lines 360-366). Namely, in SHIP:

“Five MDD-related cluster PRSs in the SHIP cohort (N = 1108) were positively correlated with their cluster membership probability; for Clusters 1 and 7 these correlations reached (suggestive) significance ($p_{cl1}=0.025$; $p_{cl7}=0.067$). The correlation pattern among these seven PRSs was similar to the correlation patterns observed at the phenotypic and genetic levels in the UKB and FinnGen cohorts (Supplementary Fig. S19).”

For THL:

“All PRSs showed a significant positive association with the cluster probability in the THL cohorts (Benjamini-Hochberg adjusted p-values range from 1.0×10^{-15} to 1.7×10^{-2} , Supplementary Fig. S14, see Supplementary Table S31 for details of explained variance by the PRS).”

These analyses were crucial for validating the applicability of our cluster definitions beyond the initial discovery cohort.

While the authors are aware of the differences between cohorts such as age-distribution, observation periods, and healthcare provision, it is unclear how the possible effects are handled. In fact, participants seem to be aligned by age (age=0 years; Fig. 1B) rather than by birth-year. This implies that MDD-individuals in the same age-bin (e.g. age 0-20 yr) may have been born and diagnosed years apart and even with different diagnostic classifications, clinical practice and healthcare provision. Smoking habits (strongly linked to MDD-liability) have changed considerably over the past decades and therefore also somatic consequences of smoking have changed.

Thank you for raising this important observation regarding handling cohort differences such as age distribution, observation periods, and healthcare provision. We recognize this as a limitation of our study, which we already mentioned briefly in the Discussion:

“Although our method was supported by temporal disease information from public health data, this approach had several limitations and involved simplification. Differences in healthcare systems lead to differences in disease rates, possibly related to differences in year of birth and age.”

In aligning participants by age (e.g., age 0-20 years) rather than by birth year, we made a deliberate methodological choice. This approach inherently means that individuals in the same age bin may have been born and diagnosed years apart, potentially under different diagnostic classifications and

healthcare contexts. While we acknowledge that this poses a valid concern, the rationale for our decision is grounded in the recognition that biological age is a pivotal determinant in many disorders. This approach, deeply embedded in our model, was selected to ensure that our analyses were primarily driven by biological factors rather than chronological ones.

Furthermore, our ability to derive clusters in three large, independent cohorts, each varying significantly in cultural, geographical, and healthcare aspects, underscores the robustness of our methodology. Despite the inherent differences among these cohorts, our approach successfully identified consistent clusters, demonstrating the universality and transferability of our findings across diverse populations. By focusing on clinical data (disease onset information) and employing relatively stable ICD-10 level-3 codes, we aimed to mitigate the impact of the variations between cohorts. This is also apparent in the similarity of the cumulative number of diseases between the three cohorts (see the newly inserted Supplementary Figure S2):

Figure S2. The cumulative count of disease diagnoses with increasing age across the three discovery cohorts (UKB, CHSS, THL).

Nevertheless, we recognize that our methodology, while effective in many respects, has its limitations due to the decision to align participants by age. Given the central role of biological age in disease patterns, we believe this decision was justified, but we also acknowledge the complexity it brings to our study's interpretation. We also understand that this choice carries the limitation of not fully

accounting for the temporal changes in clinical practices, diagnostic classifications, and healthcare contexts that can vary over the years.

Furthermore, we also give a comprehensive description of the incorporated cohorts to enhance clarity and transparency regarding the collection of temporal data in the various cohorts, detailed in Supplementary Table S1B. Moreover, we also included two figures illustrating the distribution of assigned diagnoses per year for all conditions and specifically for MDD only within the UK Biobank dataset (see Supplementary Figures S1A&B). As expected, the data density improves notably after the 1990s, aligning with the trend of increased digitalization. Nevertheless, chronic diseases diagnosed at an early age remain within the system and are captured by our data. Similarly, our dataset comprehensively covers diseases that predominantly manifest in older age, ensuring their inclusion in our analyses.

Additionally, in our response to the third reviewer's second question, we provided a detailed characterization of the weighted directed MDD-related multimorbidity scores that were used for cluster definition. Specifically, we conducted logistic regression analyses, with MDD diagnosis as the outcome and the interval-specific scores plus covariates (age, sex, income, standardized birth year) as predictors. We consistently observed a statistically significant positive relationship between the scores and MDD occurrence, indicating a significant influence of the weighted direct MDD-related multimorbidity scores on MDD risk. These relationships persisted after adjusting for the subjects' standardized birth year, indicating that the observed relationship between the scores and the occurrence of MDD is not merely a result of age-related biases or temporal trends in diagnosis practices. By including subjects' standardized birth years in the logistic regression models, the analysis controls for potential confounding effects that might arise due to variations in diagnostic criteria, healthcare access, or disease awareness over time.

The genetic analyses contrasting two clusters will by necessity be conditioned by different prevalence of early-vs-late onset disorders. More importantly, it is unclear whether the genetic analyses of all cluster-individuals inform on MDD subtypes or simply reflect differences in prevalence of other disorders in the clusters. Authors could have considered to simply estimate the SNP heritability of MDD individuals (only) across clusters to quantify a possible genetic component.

Thank you for your question. As we have stated above, our primary objective with this paper was to establish clusters across the entire population, each showing distinct trajectories of MDD-related multimorbidity. This method was deliberately chosen to allow for the inclusion of individuals in clusters whether or not they had been diagnosed with MDD, aiming for a broadly applicable clustering approach based solely on clinical data.

In response to your suggestion, we have now conducted a case-only analysis on the UKB cohort, focusing exclusively on individuals diagnosed with MDD and conducting a GWAS on these filtered

clusters in a manner similar to our approach with the full population. Through this analysis, we aimed to calculate heritability estimates and assess the genetic correlation between these MDD-specific clusters and the original clusters derived from the broader population.

Our findings revealed that the genetic correlation between the original clusters and the corresponding MDD case-only clusters was substantial, with values ranging from 0.78 to 1 (see figure below). This shows that the genetic basis of MDD patients within these clusters closely aligns with that of the entire cluster population, indicating a significant genetic similarity. Moreover, the genetic correlation patterns observed among the MDD case-only clusters were similar to those among the original clusters. However, it's important to note that the statistical power in this MDD case-only scenario was decreased due to the reduced sample size (N=28,853). The heritability estimates were also lower (1: 0.0305, 2: 0.0229, 3: 0.0196, 4: <0.0, 5: 0.0261, 6: 0.0268, 7: 0.0191) in comparison to those observed in the original clusters, as detailed in the third row of Table 2. This situation underscores the validity of our initial decision to utilize the entire population for clustering. Not only are heritability estimates higher and statistical power greater in this broader context, but it also allows us to account for individuals with undiagnosed (latent) MDD within each cluster, who would be overlooked in a case-only analysis.

Additionally, we extended our analysis to include various depression phenotypes, such as current depression score, current PHQ-9 score, current and lifetime CIDI, and clinically diagnosed MDD (F32 or F33). The genetic correlations among these depression phenotypes, as well as between them and both the original and MDD case-only clusters, showed highly similar patterns, affirming their strong genetic interrelation. (The heritability estimate for the current CIDI was so low that it precluded the computation of its genetic correlation with other traits.)

We include the new figure in the revised version of the supplementary material and indicate the results of this specific analysis in the main text (Lines 297-304):

“Moreover, we performed a UKB-specific case-only analysis focusing on individuals diagnosed with MDD, and found substantial genetic correlations (0.78-1) between the original population-based clusters and the MDD-specific clusters, underscoring a significant genetic similarity across these groups (Supplementary Figure S12). However, the reduced sample size (N=28,853) in the MDD case-only scenario led to lower heritability estimates compared to the original clusters. Extending the analysis to various depression phenotypes (Supplementary Methods) showed high genetic correlation among these, and highly similar genetic correlation patterns observed between them and the clusters.”

Figure S12. Genetic correlation between clusters, MDD case-only clusters, and various depression phenotypes. We performed a case-only analysis on the UKB cohort, focusing exclusively on individuals diagnosed with MDD and conducting a GWAS (N=28,853) on these filtered clusters using the same methodology as with the full population. The plot shows the genetic correlation computed with the LD Score Regression (LDSC) method among the original clusters, the MDD case-only clusters, and various depression phenotypes. Genetic correlation values not computable due to low heritability are denoted with a question mark. All estimated genetic correlations are bounded between -1 and +1, with values beyond this range marked by stars. Additionally, genetic correlations deemed not significant, following Benjamini-Hochberg correction with an adjusted p -value < 0.05 , are crossed out.

Reviewer #3. Psychiatry and genetics.

Reviewer #3 (Remarks to the Author):

Dear Reviewer #3,

We sincerely appreciate your time and effort in reviewing our manuscript, "Unique genetic and risk-factor profiles in multimorbidity clusters of depression-related disease trajectories from a study of 1.2 million subjects." We are grateful for your insightful comments and constructive feedback, and we anticipate that they will significantly enhance the strength and impact of our paper.

General

The authors set out to identify subtypes of MDD using information from multimorbidities in patients with MDD, identifiable in large biobanks such as UKBiobank and FinnGen. They first use Bayesian networks to identify a minimal, nonredundant set of MDD-related multimorbidity trajectories from the entire medical histories, then assess patient similarities using this filtered set of trajectories, and finally cluster patients across cohorts using a privacy preserving Bayesian approach. The paper is well-written with excellent figures illustrating the concepts and results. My experience is with MDD genetics, and I am not familiar with some of the bayesian approaches the authors have made use of, therefore my comments would be mostly on the genetics side, with some questions on the methods.

Comments

1. Identifying strongly relevant multimorbidities: I am unfamiliar with the inhomogenous dynamic bayesian network approach, but after reading the explanation in Methods, I am still unclear as to how the authors could determine the direction of arrows (direct edge from X to Y using the authors' words) pointing from diseases to each other in the system, without testing for causality (which is itself really difficult) or interactions (another difficult thing). I didn't see any tests for either, am I missing something here?

Response to Question 1:

Thank you for your question regarding the application of inhomogeneous dynamic Bayesian networks (iDBNs) and the interpretation of direct edges within this framework.

It is important to clarify that while the arrows in a Bayesian network can, in some contexts, mean causality, our study did not utilize them in this regard. Testing for causality or interactions in this complex system is indeed challenging and was not the aim of our analysis. Our goal was to use these

probabilistic relationships to identify diseases strongly relevant to MDD, thereby informing our understanding of MDD-related multimorbidity patterns.

Specifically, in our methodology, the directed arrows in the Bayesian network represent direct probabilistic relationships between diseases rather than causal links. This means that the presence of one disease (e.g., MDD) directly influences the probability distribution of another. To assess the strong relevance of each disease to MDD, we focused on the concept of the Markov Boundary. In Bayesian network theory, the Markov Boundary of a variable (in our case, MDD) is the smallest set containing all variables carrying information about MDD that cannot be obtained from any other variable. In other words, we cannot drop any variable from this set without losing information. By examining the diseases that are in the Markov Boundary of the MDD variable, we could calculate the strength of their probabilistic relationship with MDD. More specifically, within the Bayesian statistical framework employed in our study, we compute the posterior probability of each variable being within the Markov Boundary of MDD (i.e., the probability of their strong relevance with respect to MDD).

We hope this explanation clarifies the methodology and its application in our study. We acknowledge the complexity of these concepts and will endeavor to make the explanation in our manuscript more accessible to readers with varying levels of familiarity with Bayesian methods. For that purpose, we included a summary of the above explanation in the Online methods (see Lines 669-680).

2. weighted directed MDD-related multimorbidity scores: this seems to be a sum score of each individual's propensity to get MDD given their other diseases at each time t, weighted (0,1) by whether they actually had a first diagnosis of each other disease at each time t. This seems a completely new metric to me, and I would like to see more characterisation of this metric. Why is this the right metric to use? How is onset of MDD being taken into account? How does age range in the different cohorts and availability of data on disease onset affect this score? For example, do the individuals in UKB (age range 37-93) have disease records going back to when they are much younger, for identifying earlier onset diseases (seems not from Fig 2, as most disease onsets are after 30)? This is also supported by Fig S3, which shows that the onsets of diseases in the CHSS cohort (the youngest) seem earlier? As strong-relevance and this score form the basis for clustering and all further analysis, I find it hard to assess the merits of further analyses without understanding these concepts. All my comments from this point on are assuming these two metrics are well-characterised, which I think the authors ought to demonstrate.

Response to Question 2:

Thank you for your clarifying question regarding the weighted directed MDD-related multimorbidity scores, a central metric in our study.

The scores used for clustering are based on the **probability of strong relevance** of each disease with respect to MDD at specific time intervals. The concept of strong relevance has been well-characterized in our previous studies (Lautner-Csorba et al. 2012, Ungvári et al. 2012, Lautner-Csorba et al. 2013, Antal et al. 2014, Marx et al. 2017). We have utilized this measure as a means to reflect direct (non-mediated) relationships between variables more accurately. Moreover, earlier research demonstrated that Bayesian filtering of pairwise comorbidity associations, focusing on strong relevance rather than merely pairwise associations, significantly enhanced the shared molecular background between diseases (Marx et al. 2017).

This posterior probability of strong relevance is then utilized as a weight in a cumulative sum, where a disease's onset (incidence) at a certain time interval is weighted by its probability of strong relevance during that interval. This sum represents a participant's **weighted MDD-related multimorbidity burden**, essentially indicating the expected count of MDD-related comorbid diseases at a given time interval. It is a simplified yet natural metric for depicting an individual's multimorbidity burden. Finally, based on the temporal trajectory of the weighted MDD-related multimorbidity burden scores, we clustered all participants into seven distinct clusters.

Ideally, using the full posterior of the Markov Boundary sets of MDD would be more theoretically sound. However, we opted for using only marginals (pairwise strong relevance) for weighting disease presence due to computational constraints. While simplified, this approach proved feasible and effective in our study, as evidenced by the significant variation in the presence and onset of MDD (which was not used in defining the clusters) across different clusters. Although different analytical approaches could be employed, our study aimed to demonstrate the feasibility and effectiveness of our specific methodology.

As previously stated, the onset of MDD was not considered in the derivation of the clusters. Instead, this information was utilized post-clustering to characterize and delineate the differences among the clusters in terms of MDD. This involved a comparative analysis of both the prevalence and onset of MDD across the clusters, as illustrated in Figure 3 B&C.

To elucidate the relationship between the scores used for clustering and the incidence of MDD, we visualized the score distributions among participants in UKB, distinguishing between those diagnosed with MDD and those not diagnosed, across each time interval (see Figure SM1.A below). Notably, the score distributions were consistently and statistically significantly elevated in the patient group diagnosed with MDD compared to those without MDD diagnosis. Additionally, we conducted logistic regression analyses for each time interval, using the MDD diagnosis as the dependent variable and the interval-specific score, along with covariates such as age, sex, income, and standardized birth year, as predictors (see Figure SM1.B below). The results uniformly indicated a statistically significant and robust positive effect of the scores on MDD occurrence in every time interval. These findings affirm that the weighted direct MDD-related multimorbidity scores employed in the clustering have a

substantial impact on the likelihood of MDD occurrence. This significance holds true even when accounting for the chronological year of diagnosis, as evidenced by including the subjects' standardized birth years in the logistic regression models.

Figure SM1. Characterization of the weighted directed MDD-related multimorbidity score in the UKB. (A) Distribution of the weighted directed MDD-related multimorbidity score among individuals diagnosed and not diagnosed with MDD. (B) Results of logistic regression analyses for each time interval, using the MDD diagnosis as the dependent variable and the interval-specific score (with a prefix of “wdMDDdb”), along with covariates (age, sex, income, and standardized birth year) as predictors. 95% confidence intervals of the coefficient are indicated with lines.

To further explore the relationship between the age of MDD onset and the scores used for clustering, we generated two-dimensional density plots based on the subset of patients diagnosed with MDD. These plots juxtapose the age at MDD onset against the values of the weighted direct MDD-related multimorbidity scores at various time intervals (see figure below). The plots reveal a clear age-related trend: higher scores at earlier intervals (e.g., [0-20 years]) are associated with an earlier onset of MDD, whereas higher scores at later intervals (e.g., [0-70 years]) correspond to a later onset of MDD. This observation suggests that the scores also effectively encapsulate information about the age of MDD onset.

Figure SM2. The relationship between the age of MDD onset and the weighted direct MDD-related multimorbidity score. Each box shows a two-dimensional density plot of patients diagnosed with MDD corresponding to a certain

time interval, where the x-axis represents the age at MDD onset, and the y-axis corresponds to the weighted directed MDD-related multimorbidity score.

We include these explanations and figures in the revised version of the Supplementary methods.

Additionally, it's important to note that the biobanks comprehensively contain disease diagnoses across the known lifespan of the individuals, starting from their early years. (We give a comprehensive description of the collection of temporal data in the various cohorts in the new Supplementary Table S1B. We also included two figures illustrating the distribution of assigned diagnoses per year for all conditions and specifically for MDD only within the UK Biobank dataset, see Supplementary Figures S1A&B). As expected, the data density improves notably after the 1990s, aligning with the trend of increased digitalization. Nevertheless, chronic diseases diagnosed at an early age remain within the system and are captured by our data. Similarly, our dataset comprehensively covers diseases that predominantly manifest in older age, ensuring their inclusion in our analyses.

Figure S1. The distribution of assigned diagnoses per year within the UK Biobank dataset for all conditions (A), and specifically for MDD (B).

Moreover, we determined the cumulative count of disease diagnoses with increasing age across all three discovery cohorts and observed highly similar trends (see Supplementary Figure S2). This similarity suggests that the cohorts effectively capture comparable age-related multimorbidity patterns.

Figure S2. The cumulative count of disease diagnoses with increasing age across the three discovery cohorts (UKB, CHSS, THL).

References:

- 1.) Lautner-Csorba O, Gézsi A, Semsei ÁF, et al. (2012) Candidate gene association study in pediatric acute lymphoblastic leukemia evaluated by Bayesian network based Bayesian multilevel analysis of relevance. *BMC Med Genomics* 5, 42. <https://doi.org/10.1186/1755-8794-5-42>
- 2.) Ungvári I, Hullám G, Antal P, et al. (2012) Evaluation of a Partial Genome Screening of Two Asthma Susceptibility Regions Using Bayesian Network Based Bayesian Multilevel Analysis of Relevance. *PLoS ONE* 7(3): e33573. <https://doi.org/10.1371/journal.pone.0033573>
- 3.) Lautner-Csorba O, Gézsi A, Erdélyi DJ, Hullám G, Antal P, et al. (2013) Roles of Genetic Polymorphisms in the Folate Pathway in Childhood Acute Lymphoblastic Leukemia Evaluated by Bayesian Relevance and Effect Size Analysis. *PLoS ONE* 8(8): e69843. <https://doi.org/10.1371/journal.pone.0069843>
- 4.) Antal P, Millinghoffer A, Hullám G, et al. (2014) Bayesian, systems-based, multilevel analysis of associations for complex phenotypes: from interpretation to decisions. *Probabilistic Graphical Models for Genetics, Genomics and Postgenomics*. Oxford University Press: Oxford, UK.
- 5.) Marx P, Antal P, Bolgar B, Bagdy G, Deakin B, Juhasz G (2017) Comorbidities in the diseasome are more apparent than real: What Bayesian filtering reveals about the comorbidities of depression. *PLoS Comput Biol* 13(6): e1005487. <https://doi.org/10.1371/journal.pcbi.1005487>

3. GWAS on clusters: something is going on in chromosome 6 across GWAS on all clusters - the fact the entire chromosome is lighting up seems an impossible real finding to me. The authors wrote about immune profiles in the main text, but does not address whether this signal is likely an artefact. In Fig S15, the authors showed the GWAS performed in FinnGen and I don't see the entire chromosome 6 lighting up. Table 3 further showed that results from FinnGen mostly do not replicate findings from the UKB analysis (<50% for loci and <15% at gene level in all clusters, though the authors wrote successful replication in the main text). r_G between the UKB clusters and FinnGen clusters are also moderate at best (I would like to see heritability estimates also), and the authors didn't give values of variance explained by PRS from UKB on the smaller cohorts (only the p values, from which I can't tell how good the predictions are). Also note the replicated genes are also mostly in the MHC region, which is highly influenced by population structure and therefore potentially artefacts.

Response to Question 3:

In response to your comment regarding the chromosome 6 anomaly in our GWAS analysis across all clusters, we have taken your observation into serious consideration and conducted a detailed investigation to ascertain the source of what appeared to be artefacts. Our inquiry led us to identify a specific issue within our GWAS QC pipeline, particularly related to a processing step executed by plink. This issue was communicated to the author of plink, which subsequently resulted in a significant bug fix as announced on the plink website (see <https://www.cog-genomics.org/plink/2.0/>):

"5 Jan: Fixed multithreaded --make-bed data corruption bug that could occur when writing >260k samples while filtering or reordering samples, or writing >520k samples without filtering/reordering. [...] Data corruption from this bug should be pretty obvious, but if you were in the habit of using --bfile/--make-bed instead of --pfile/--make-pgen with UK Biobank data, you may want to rerun large --make-bed operations and downstream steps to be safe."

Following the release of the bug-fixed version of plink, we reanalyzed all UK Biobank GWAS data from scratch. This reanalysis yielded results that were not only similar to our initial findings but also demonstrated even stronger associations. Our initial QC process had effectively eliminated the majority of corrupted SNPs, yet a small fraction (<0.1%) managed to evade detection, contributing to the unusual patterns observed in our initial Manhattan plots. The updated GWAS QC, leveraging the corrected version of plink, increased the number of SNPs included in our analysis by 60% (from 3.88 million to 6.27 million). Sample size changed from 247,325 to 249,167. Changes in the number of significant loci, SNPs, genes, gene sets, and MDD-overlapping genes, following this reanalysis, are detailed in the revised version of Table 2.

This comprehensive reanalysis also led to minor variations in enriched gene sets and functional analyses (Lines 249-258), alongside an enhanced alignment with the MDD analysis by Howard et al., marked by a higher count of pleiotropic genes and gene modules (see last row of Table 2 and Lines 310-324).

Furthermore, the concordance between our findings and those from the FinnGen study improved significantly, as evidenced by an increase in the percentage of SNPs showing the same direction of effect across several clusters and a rise in the number of replicated loci (see revised Table 3). Although the statistical power of the UKB analysis was increased, leading to a greater number of significant and replicated genes across clusters, the proportion of replicated genes relative to significant genes in the UKB analysis saw a slight decline. Besides, following the reanalysis, we observed an increased concordance between the Polygenic Risk Scores (PRS) and cluster membership within the THL cohort. Additionally, we have detailed the variance explained by the PRS for the target variable in a newly added supplementary table (see Supplementary Table S31).

In response to your comment regarding the replication of genes predominantly in the MHC region, we note that after the reanalysis two-thirds (16 out of 24) of the replicated genes are located outside of the MHC region (see Supplementary Table S20B). Besides, to address potential concerns about population structure, we explicitly controlled for it in each GWAS conducted for every cohort. This was achieved by including the top 10 principal components as covariates in the regression analysis and focusing our analyses on the Caucasian population. Additionally, we recalculated the genetic correlation between the clusters after excluding the MHC region from our analysis. The results agreed with the original (without excluding MHC) analysis (see Figure 4B), accurate up to two significant digits. This supports the robustness of our findings, affirming that the observed genetic associations extend beyond the MHC region and are not artifacts of population structure.

Overall

The authors showed a high level of sophistication in their work that I feel is not matched by rigour to characterise new metrics they are proposing and findings that seem to harbour artefacts. I am not currently convinced by their methods and findings, but I am happy to be proven wrong if they are able to demonstrate that the metrics are well-suited for disease trajectory characterisation, and their clusters aren't just tagging population structure.

REVIEWER COMMENTS

Reviewer #1 (Remarks to the Author):

I appreciate the extensive response of the authors and they have fully addressed my concerns raised in the initial review. I have no further questions or suggestions.

Reviewer #2 (Remarks to the Author):

I greatly appreciate the careful and elaborate replies and discussions provided by the authors in response to the reviews of their study resulting in a much-improved manuscript. Still, I do find the manuscript remains somewhat hard to follow, in my opinion for two reasons.

First, the logic, narrative and the language still seem dominated by a computational mindset that in my opinion (a) complicated the original manuscript, (b) prompted three troubled reviews based on what seemed to be quite different readings of the reviewers, and (c) necessitated very elaborate replies. I suggest that the authors provide a table/figure that not only elaborate on Figure 1a (pipeline) but in parallel explains and discuss the rationale underlying the analytical strategy.

Second, important medical, psychiatric, and epidemiological context is lacking, notably:

(a) Alignment with solid population-based knowledge of type and temporal characteristics of depression comorbidities (e.g. prevalence and age-at-onset established primarily in the Nordic countries). Simple comparisons of the frequencies of MDD-relevant disorders across the cohorts and with external population-based estimates would be reassuring.

(b) Classical biases such as truncation and censoring of disease information on participants should be acknowledged and possible impact discussed; quite trivially, early onset disorders are unlikely to be recorded for older participants while younger participants will not have developed late-onset illnesses raising (at least) the possibility that some clusters arise or are affected by these biases. Another cause of biases is the well-known differences between healthcare provision across countries.

(c) I do appreciate the authors' choice to pursue a data-driven approach with respect to selection of depression-relevant diagnoses, but not even Bayesian approaches can correct biased input data. The early age of individuals in cluster 1-2 relative to individuals in clusters 3-7, while presented as an interesting and even important finding, may simply be a consequence of different truncation and censoring across the cohorts.

Third, I am worried that the authors do not consider or at least discuss that the GWASes contrast clusters (of individuals) that by construction have defining differences in the prevalence of somatic and psychiatric disorders (except depression), and thus by construction are expected to identify gene variants already known to associate with these pre-selected disorders. Another observation, presented as pleiotropy at the gene level, is the genes implicated by the most recent MDD-GWAS and genes significantly associated with one or more of the seven clusters; however, the null hypothesis is not mentioned, and the reader is left with the impression that these are simple observations (no reference to Methods section).

In conclusion, while I recognize the creative analytical approach, the many considerations, and the considerable effort in carrying out the elaborate set of analyses; I remain skeptical of the underlying biological/clinical rationale on which these analyses rest. The root cause of this skepticism is in part the absence of well thought-through and articulated hypotheses that would have allowed the reader to appreciate the medical / biological rationale and evaluate the null as the analyses are being presented and conducted; the authors typically express what they hope to find, rather than presenting the hypothesized driving mechanism that is being tested. Biological observations – by virtue of being biological – are correlated; the challenge is not to note such correlations, but to understand which are 'directly' causal and biologically or clinically important.

Reviewer #3 (Remarks to the Author):

I am satisfied with the authors responses

Reviewer #4 (Remarks to the Author):

The authors utilized dynamic Bayesian network approach to identify MDD subtypes. They calculated posterior probability of each disease's strong relevance with respect to MDD. Such probabilities are used to form a Markov boundary of the MDD variables which include 86 MDD-related disorders. For each participant, a 4-dimensional vector was constructed where each coordinate corresponds to the weighted direct MDD-related multimorbidity scores for each of the 4 time intervals. Consequently, using the k-means clustering algorithm, they identified seven clusters of disease-burden trajectories throughout the lifespan among 1.2 million participants.

1. Line 681, the posterior probabilities were estimated using MCMC simulations. I'm sure the authors have checked the convergent criteria for their simulations. It would be better to provide, at least in the supplementary methods, some details regarding the convergence of the chains.

2. Line 705, the authors selected diseases that had a posterior probability of strong relevance with respect to MDD >0.5 in at least one time interval for at least one cohort. Such criterion will include the diseases that had the probability >0.5 in exactly one time interval for exactly one cohort. Inclusion of such diseases will not support the authors' claim of "each disease is consistently represented across all cohorts" or "guaranteeing their universal presence across the diverse cohort datasets". My suggestion is to replace the criterion of "at least one" with "more than one", i.e. the selected diseases had a posterior probability of strong relevance with respect to MDD >0.5 in more than one time interval for more than one cohort. This will ensure that the selected diseases are present consistently in time and universal across all cohort.

3. Line 744, the s_j should be c_j .

Reviewer #5 (Remarks to the Author):

This is a very comprehensive and large-scale study on major depression. The study employs advanced statistical techniques and provides important clinical insights into possible subtypes of MDD, considering its trajectory and genetic and non-genetic factors. Overall, the paper is also well-written and explains the methods pretty well. Here are some comments for consideration (I shall mainly focus on the methods and presentation)

- P31 top – I understand A and B can interact to cause C (but A or B alone may not cause C), or say B enhances the effect of A on the outcome C.

Perhaps I miss something, could the authors explain whether/how this interaction is directly modeled in your Bayesian network or via other methods? I guess it may not be very straightforward to model effect modification by a 3rd variable with standard BN methods?

- Could you discuss further the assumptions of the DBN (and other methods)? For example, do we need to assume lack of unknown confounders, lack of selection bias etc.?

- Line 671 p 31 – if "presence of one disease (e.g., MDD) directly influences the probability distribution of another" -> in my understanding, this is very close to being causal, although the authors say this is not. Could you further clarify this concept?

- The authors mentioned privacy-preserving Bayesian federated clustering as a methodology novelty. Could they further describe it in methods and highlight how it can be achieved?

- The authors presented or highlighted some sig genes in table 2, and possibly some other loci are not listed there. Could they discuss whether the differences in the sig. genes link up to the different disease profiles in the 7 clusters? Eg some genes are specifically related to certain diseases in a cluster.

- About software availability: May I know if the BN-BMLA software (for BN building) is publicly available as a package? If so, please list the link as well.

Unique genetic and risk-factor profiles in multimorbidity clusters of depression-related disease trajectories from a study of 1.2 million subjects

NCOMMS-23-36313A

REVIEWER COMMENTS

Reviewer #1 (Remarks to the Author):

I appreciate the extensive response of the authors and they have fully addressed my concerns raised in the initial review. I have no further questions or suggestions.

Dear Reviewer,

We sincerely appreciate your time, effort, and thoughtful consideration in reviewing our revisions and accepting them.

Reviewer #2 (Remarks to the Author):

I greatly appreciate the careful and elaborate replies and discussions provided by the authors in response to the reviews of their study resulting in a much-improved manuscript. Still, I do find the manuscript remains somewhat hard to follow, in my opinion for two reasons.

First, the logic, narrative and the language still seem dominated by a computational mindset that in my opinion (a) complicated the original manuscript, (b) prompted three troubled reviews based on what seemed to be quite different readings of the reviewers, and (c) necessitated very elaborate replies. I suggest that the authors provide a table/figure that not only elaborate on Figure 1a (pipeline) but in parallel explains and discuss the rationale underlying the analytical strategy.

Dear Reviewer,

We sincerely appreciate your encompassing and thoughtful consideration in reviewing our revisions.

According to your suggestion, we updated Fig. 1 with a panel explaining the biological/medical rationale of the study and formulating our main hypothesis. We also added a sentence to the introduction stating our main hypothesis (Lines 75-78), which was further elaborated in the results section at the Dynamic Bayesian network analysis reveals seven MDD-related multimorbidity clusters section (lines 133-141).

Introduction (Lines 75-78)

“Our main hypothesis was that the use of age-dependent strongly relevant MDD-related multimorbidities enrich the genetic basis of MDD, such that specific participant clusters are associated with distinct genetic profiles contributing to the pathology of major depressive disorder.”

Second, important medical, psychiatric, and epidemiological context is lacking, notably:

(a) Alignment with solid population-based knowledge of type and temporal characteristics of depression comorbidities (e.g. prevalence and age-at-onset established primarily in the Nordic countries). Simple comparisons of the frequencies of MDD-relevant disorders across the cohorts and with external population-bases estimates would be reassuring.

Thank you for this suggestion. The age at onset and prevalence of MDD-relevant disorders across the cohorts can be found in Supplementary Table S2, where we can see cohort-specific differences, probably due to distinct healthcare practices and different cohort ascertainments (see Online Methods section, lines 519-648); however, the main patterns are similar to previously published epidemiologic data. Furthermore, now in Fig. 1 we summarise previous medical, psychiatric, and epidemiological observations that led to our hypothesis. Namely, accumulating evidence suggested that MDD is frequently comorbid not only with other psychiatric disorders but also with several somatic diseases contributing to worse health-related outcomes and decreasing quality of life (Merikangas and Kalaydjian 2007, Moussavi et al. 2007, Berk et al. 2023, Paskan et al. 2023). Thanks to network medicine and system biology approaches, it has been demonstrated that comorbid conditions partially represent common biological mechanisms (Barabási et al. 2011, Menche et al. 2015, Wang et al. 2017, Brainstorm Consortium 2018). Furthermore, directly related comorbidities of depression, where the relationships are not mediated by other disorders, represent stronger molecular-level relationships (Marx et al. 2017) and are time-dependent (i.e., vary with onset age, Rzhetsky et al. 2007). Finally, a recent comorbidity mapping study of asthma supported that comorbidities are indeed suitable for delineating distinct subgroups (so-called endotypes) of complex multifactorial disorders (Jia et al. 2022).

References

Merikangas KR and Kalaydjian A. Magnitude and impact of comorbidity of mental disorders from epidemiologic surveys. Curr Opin Psychiatry 20:353–358. 2007 PMID: 17551350; DOI: 10.1097/YCO.0b013e3281c61dc5

Moussavi S et al.. Depression, chronic diseases, and decrements in health: results from the World Health Surveys. Lancet 2007 Sep 8;370(9590):851-8. PMID: 17826170; DOI: 10.1016/S0140-6736(07)61415-9

Berk M et al. Comorbidity between major depressive disorder and physical diseases: a comprehensive review of epidemiology, mechanisms and management. World Psychiatry 2023 Oct;22(3):366-387. PMID: 37713568; DOI 10.1002/wps.21110

Pasman JA et al. Epidemiological overview of major depressive disorder in Scandinavia using nationwide registers. *Lancet Reg Health Eur.* 2023 Mar 28;29:100621. PMID: 37265784; DOI: 10.1016/j.lanep.2023.100621

Barabási AL et al. Network medicine: a network-based approach to human disease. *Nat Rev Genet.* 2011 Jan;12(1):56-68. PMID: 21164525; DOI: 10.1038/nrg2918

Menche J et al. Uncovering disease-disease relationships through the incomplete interactome. *Science* 347, 1257601 (2015). DOI: 10.1126/science.1257601

Wang K et al. Classification of common human diseases derived from shared genetic and environmental determinants. *Nat Genet.* 2017 Sep;49(9):1319-1325. PMID: 28783162; DOI: 10.1038/ng.3931

Brainstorm Consortium. Analysis of shared heritability in common disorders of the brain. *Science.* 2018 Jun 22;360(6395):eaap8757. PMID: 29930110; DOI: 10.1126/science.aap8757

Marx P et al. Comorbidities in the diseasome are more apparent than real: What Bayesian filtering reveals about the comorbidities of depression. *PLoS Comput Biol* 2017 Jun 23;13(6):e1005487. eCollection 2017 Jun. PMID: 28644851; DOI: 10.1371/journal.pcbi.1005487

Rzhetsky A et al. Probing genetic overlap among complex human phenotypes. *Proc Natl Acad Sci U S A.* 2007 Jul 10;104(28):11694-9. PMID: 17609372; DOI: 10.1073/pnas.0704820104

Jia G et al. Discerning asthma endotypes through comorbidity mapping. *Nat Commun* 2022 Nov 7;13(1):6712. doi: 10.1038/s41467-022-33628-8. PMID: 36344522

(b) Classical biases such as truncation and censoring of disease information on participants should be acknowledged and possible impact discussed; quite trivially, early onset disorders are unlikely to be recorded for older participants while younger participants will not have developed late-onset illnesses raising (at least) the possibility that some clusters arise or at affected by these biases. Another cause of biases is the well-known differences between healthcare provision across countries.

Thank you for your comments on potential biases such as truncation and censoring of disease information in our dataset. We acknowledge these concerns and have implemented several strategies in our study design to mitigate their impact.

Firstly, it's important to clarify that there is no truncation of disease onset data in our study for the UK Biobank and THL cohorts, as all past disease information was recorded at the time of assessment, ensuring comprehensive historical data coverage. Similarly, the CHSS cohort's documentation of disease onset extends back to 1913. Meanwhile, we recognize that the coverage

of the disease onset data is not uniform across all cohorts and all time intervals, as detailed in the Methods section of our paper (see lines 519-648).

Secondly, our Dynamic Bayesian Network (DBN) methodology plays a critical role in addressing potential censoring issues. The DBN accounts for censoring by including only those participants in each cumulative time interval who have complete disease onset information up to the end of that interval (see Fig. 1B and Online Methods, lines 722-728). This ensures that our analyses in each time slice are based on complete records. Additionally, our cluster definitions are specifically based on participants who have complete onset information up until the age of 70 years (see lines 769-772). Participants who do not meet this criterion are assigned to clusters based on the available data (see lines 772-775).

Regarding the variability in healthcare provision across different countries, this is indeed a known limitation of our model. We have discussed this issue in depth in our previous review, noting how these differences might affect the generalizability of our findings. While these healthcare disparities are challenging to completely control for, as we emphasize this limitation in the paper (see lines 501-503), they are important in interpreting our results within a broader epidemiological context.

(c) I do appreciate the authors choice to pursue a data-driven approach with respect to selection of depression-relevant diagnoses, but not even Bayesian approaches can correct biased input data. The early age of individuals in cluster 1-2 relative to individuals in clusters 3-7, while presented as an interesting and even important finding, may simply be a consequence of different truncation and censoring across the cohorts.

The design of the study is the same for all cohorts, i.e., we mitigated the biases of censoring, and the truncation is not present.

Third, I am worried that the authors do not consider or at least discuss that the GWASes contrast clusters (of individuals) that by construction have defining differences in the prevalence of somatic and psychiatric disorders (except depression), and thus by construction are expected to identify gene variants already known to associate with these pre-selected disorders. Another observation, presented as pleiotropy at the gene level, is the genes implicated by the most recent MDD-GWAS and genes significantly associated with one or more of the seven clusters; however, the null hypothesis is not mentioned, and the reader is left with the impression that these are simple observations (no reference to Methods section).

Thank you for your comment. Our clarified hypothesis now states that the multimorbidities strongly relevant to MDD partially share its molecular background, suggesting that these disorders enrich genes that exhibit pleiotropy with respect to MDD. Importantly, the clusters are designed to reflect

varying temporal trajectories of MDD-related multimorbidity burden over the lifespan rather than on the presence or prevalence of individual comorbid diseases. Each cluster represents an aggregated profile of multimorbidity, focusing on the dynamic course of MDD-related diseases. The differences in disease prevalences observed among clusters stem directly from these distinct temporal trajectories. Consequently, each cluster inherently enriches specific pleiotropic genes that share a common molecular basis with MDD and influence the particular pattern of multimorbidity trajectory that defines the cluster. In short, this approach is designed to identify those genetic variants that are not merely associated with individual comorbid conditions but are integral to the broader biological context of MDD-related multimorbidity trajectories. In other words, our work investigates the fundamental open question of the role of pleiotropy in MDD, for example, related to somatic diseases and a wide range of brain-body interactions.

We have demonstrated these genetic correlations between diseases and MDD-related clusters in Supplementary Fig. S13, and they are further discussed in the Results section (lines 318-322). This observation aligns with findings from network medicine and genetic association studies that highlight common biological mechanisms among comorbid disorders (Barabási et al. 2011, Menche et al 2015, Wang et al 2017, Brainstorm Consortium 2018). We have now illustrated this mechanism in Fig. 1, Rationale panel.

In addition to the above, we performed further analyses to validate the main hypothesis of the study, which demonstrated significant enrichment of MDD-associated genes within the clusters.

We included these results in the revised version of the paper (Lines 328-333):

"Our results demonstrate significant enrichment of MDD-associated genes within the clusters, validating our hypothesis that strongly relevant MDD-related multimorbidities enhance the genetic background of MDD. According to the hypergeometric test, significant overlap was observed between MDD genes and cluster-specific genes in three clusters. Additionally, gene set enrichment analysis revealed that MDD genes were consistently and significantly enriched across the ranked list of cluster-specific genes in all seven clusters."

Previously, we also indicated that we identified several network modules (at least one for each cluster) that significantly enriched the signals of MDD-associated genes (see lines 334-341).

References:

Barabási AL et al. Network medicine: a network-based approach to human disease. Nat Rev Genet. 2011 Jan;12(1):56-68. PMID: 21164525; DOI: 10.1038/nrg2918

Menche J et al. Uncovering disease-disease relationships through the incomplete interactome. Science 347, 1257601 (2015). DOI: 10.1126/science.1257601

Wang K et al. Classification of common human diseases derived from shared genetic and environmental determinants. Nat Genet. 2017 Sep;49(9):1319-1325. PMID: 28783162; DOI: 10.1038/ng.3931

Brainstorm Consortium. Analysis of shared heritability in common disorders of the brain. *Science*. 2018 Jun 22;360(6395):eaap8757. PMID: 29930110; DOI: 10.1126/science.aap8757

In conclusion, while I recognize the creative analytical approach, the many considerations, and the considerable effort in carrying out the elaborate set of analyses; I remain skeptical of the underlying biological/clinical rationale on which these analyses rests. The root cause of this skepticism is in part the absence of well thought-through and articulated hypotheses that would have allowed the reader to appreciate the medical / biological rationale and evaluate the null as the analyses are being presented and conducted; the authors typically express what they hope to find, rather than presenting the hypothesized driving mechanism that is being tested. Biological observations – by virtue of being biological – are correlated; the challenge is not to note such correlations, but to understand which are ‘directly’ causal and biologically or clinically important.

We deeply share the view about the importance of filtering non-direct, mediated correlations and understanding the biological mechanisms of substantial relations. In fact, the current work is a temporal, cross-cohort extension of our earlier work with the title “Comorbidities in the diseasome are more apparent than real: What Bayesian filtering reveals about the comorbidities of depression” (see Marx et al. 2017). Along this line, we also carried out other studies to evaluate our approach that are published now and will be added to our paper as supporting evidence.

First, we tested the performance of the filtered strongly relevant MDD-related multimorbidities predicting their impact on individuals and healthcare systems, and multimorbidity progression. By creating a Multimorbidity Adjusted Disability Score (MADS) we successfully predicted health risks (e.g. incidence of new-onset depression-related illnesses, disease progression, mortality) and utilization of healthcare resources (e.g. pharmacological and non-pharmacological healthcare expenditures, González-Colom R et al. 2024).

Second, we analyzed the gene x childhood maltreatment interaction on the 7 MDD-related multimorbidity clusters with a special focus on 31 candidate genes that were previously implicated in the depressogenic effect of childhood maltreatment. Our results demonstrated that previous candidate gene results can be replicated in a cluster-specific manner (candidate genes CREB1, DBH, and MTHFR - Cluster 5; TPH1 - Cluster 6 survived multiple testing correction; Bonk S et al. 2024), further supporting that our clusters represent specific biological mechanisms.

However, it is important to note that direct disease relationships do not necessarily mean causal relationships, which we clarified further in the Online Methods (Lines 703-714).

“We also note that while the relationships in our Bayesian network are direct and unmediated by other diseases, they do not necessarily imply causation. This directness refers to the absence of intermediate variables within the network’s model structure, distinguishing these relationships from mere correlations at the abstraction level defined by the entire set of variables in the analysis. However, direct probabilistic relationships in the Bayesian network are derived from observational

data, not from interventional studies that manipulate one variable to directly observe its effect on another. Without the ability to control or manipulate the conditions, the relationships might still be influenced by unobserved confounding factors. The direct relationships in the network are based on the strongest statistical dependencies observed in the data, but these dependencies alone do not fulfil all criteria required to establish causality, such as eliminating all potential confounders and demonstrating that the relationship is not reversible."

Although this is out of the scope of the current study, our work could form the basis of future investigation to test the direct and causal effects of the identified clusters. For example, Mendelian randomization could offer a proper candidate methodology for this purpose.

References:

Marx P et al. Comorbidities in the diseasome are more apparent than real: What Bayesian filtering reveals about the comorbidities of depression. PLoS Comput Biol 2017 Jun 23;13(6):e1005487. eCollection 2017 Jun. PMID: 28644851; DOI: 10.1371/journal.pcbi.1005487

González-Colom R et al. Multicentric Assessment of a Multimorbidity Adjusted Disability Score to stratify depression-related risks using temporal disease maps. Journal of Medical Internet Research. 23/05/2024:53162 (forthcoming/in press) DOI: 10.2196/53162

Bonk S et al. Impact of gene-by-trauma interaction in MDD-related multimorbidity clusters. J Affect Disord. 2024 Aug 15;359:382-391. doi: 10.1016/j.jad.2024.05.126. Epub 2024 May 26. PMID: 38806065.

Reviewer #3 (Remarks to the Author):

I am satisfied with the authors responses

Dear Reviewer,

We sincerely appreciate your time, effort, and thoughtful consideration in reviewing our revisions and accepting them.

Reviewer #4 (Remarks to the Author):

Dear Reviewer #4,

We sincerely appreciate your time and effort in reviewing our manuscript, "Unique genetic and risk-factor profiles in multimorbidity clusters of depression-related disease trajectories from a study of 1.2 million subjects." We are grateful for your insightful comments and constructive feedback, and we anticipate that they will significantly enhance the strength and impact of our paper.

The authors utilized dynamic Bayesian network approach to identify MDD subtypes. They calculated posterior probability of each disease's strong relevance with respect to MDD. Such probabilities are used to form a Markov boundary of the MDD variables which include 86 MDD-related disorders. For each participant, a 4-dimensional vector was constructed where each coordinate corresponds to the weighted direct MDD-related multimorbidity scores for each of the 4 time intervals. Consequently, using the k-means clustering algorithm, they identified seven clusters of disease-burden trajectories throughout the lifespan among 1.2 million participants.

1. Line 681, the posterior probabilities were estimated using MCMC simulations. I'm sure the authors have checked the convergent criteria for their simulations. It would be better to provide, at least in the supplementary methods, some details regarding the convergence of the chains.

We have checked the convergence of the MCMC simulations using Geweke scores and included the following results in the revised version of our manuscript (see section "Identification of MDD-related clusters based on disease trajectories" in the Online methods, Lines 718-721):

"Convergence diagnostic testing using Geweke scores indicated that the MCMC chains had converged for 618 out of 621 (99.5%) of the posterior probabilities of the variables' strong relevance, with their z-scores within the acceptable range of -2 to 2, suggesting overall convergence of the chains."

2. Line 705, the authors selected diseases that had a posterior probability of strong relevance with respect to MDD >0.5 in at least one time interval for at least one cohort. Such criterion will include the diseases that had the probability >0.5 in exactly one time interval for exactly one cohort. Inclusion of such diseases will not support the authors' claim of "each disease is consistently represented across all cohorts" or "guaranteeing their universal presence across the diverse cohort datasets". My suggestion is to replace the criterion of "at least one" with "more than one", i.e. the selected diseases had a posterior probability of strong relevance with respect to MDD >0.5 in more than one time interval for more than one cohort. This will ensure that the selected diseases are present consistently in time and universal across all cohort.

Thank you for your suggestion regarding the selection criteria for diseases based on their posterior probability of strong relevance with respect to MDD. Your proposed adjustment to the selection criteria, aiming for "more than one" instead of "at least one," is indeed stricter and would lead to a more conservative selection of diseases.

We explored implementing the revised criterion in two different ways, both of which were consistent with the stricter definition. In the first method, we first selected intervals for each disease that were relevant in more than one cohort and then selected diseases that were relevant in more than one interval; in the second method, we evaluated diseases across intervals within each cohort, choosing

those that showed relevance in at least two intervals in more than one cohort. These methods identified only 10 and 14 universally relevant diseases, respectively.

However, it's important to clarify that the criterion of "at least one" was utilized only during a preliminary filtering step, in which we aimed to gather the broadest possible set of cross-cohort diseases. We refer to these variables in the paper as the 86 cross-cohort disease categories. When we stated that "each disease is consistently represented across all cohorts," we intended to convey that each disease variable is present in all datasets, not necessarily that it is universally or equally relevant across all analytical contexts.

After this prefiltering, we conducted an additional round of relevance analysis within each cohort, limiting variables to this broad but potentially relevant set of variables. In calculating the weighted direct MDD-related multimorbidity scores, we used relevance scores – averaged across all three cohorts – as weights. This approach ensured that each disease's influence on our final analysis was proportionate to its relevance score. In other words, the method guarantees that diseases with lower relevance have less impact on the final model, even if they pass the initial filter.

We have clarified this aspect further in the revised version of the manuscript (see section "Cross-cohort disease categories and relevance scores" in the Online methods, Lines 744-749).

3. Line 744, the s_j should be c_j .

Thank you for pointing out the error in the formula. We have corrected it in the revised manuscript.

Reviewer #5 (Remarks to the Author):

This is a very comprehensive and large-scale study on major depression. The study employs advanced statistical techniques and provides important clinical insights into possible subtypes of MDD, considering its trajectory and genetic and non-genetic factors. Overall, the paper is also well-written and explains the methods pretty well. Here are some comments for consideration (I shall mainly focus on the methods and presentation)

- P31 top – I understand A and B can interact to cause C (but A or B alone may not cause C), or say B enhances the effect of A on the outcome C.

Perhaps I miss something, could the authors explain whether/how this interaction is directly modeled in your Bayesian network or via other methods? I guess it may not be very straightforward to model effect modification by a 3rd variable with standard BN methods?

In our Bayesian network framework, we primarily focus on capturing structural probabilistic relationships between variables rather than quantifying interaction terms that occur in regression models. Although these interactions are primarily quantified at the parametric level—through the

conditional probability distributions of variables given others—our analyses primarily aimed to elucidate the structural relationships by performing exact Bayesian averaging over the parametric level rather than quantifying these interaction effects directly.

This structural approach is pivotal in observing how variables might indirectly influence each other through a common outcome. Consider a scenario where two variables, A and B, are independent but both influence a third variable C, or where B modulates the effect of A on C, creating a typical v-structure in the graph: $A \rightarrow C \leftarrow B$. However, the same v-structure could also appear if A and B are independent, but B and C interact to influence A. In such models, even though A and B are independent, their relationship becomes conditionally dependent when the value of C is known. Thus, knowing C, the value of B provides information about A.

This conditional dependence is central to understanding how interactions can be inferred within our network. Specifically, we focused on estimating the posterior probability that a variable is directly relevant to MDD or participates in interaction effects concerning MDD. This involves determining whether a variable is within the Markov Boundary of MDD, indicating strong relevance between that variable and our target variable (i.e., strong relevance is a symmetric relationship - if and only if A is in the Markov Boundary of B, then B is in the Markov Boundary of A).

In short, our current methodology centers on identifying and interpreting these structural relationships; meanwhile, it does not report quantitative interaction effects in the manner that traditional statistical models might.

- Could you discuss further the assumptions of the DBN (and other methods)? For example, do we need to assume lack of unknown confounders, lack of selection bias etc.?

The basic assumption of Bayesian networks is that independencies are the central elements in scientific and engineering modeling, so this representation is usually used as a sound but not complete representation of independencies. Of course, any assumption to exclude confounders and cryptic relatedness, constraints, etc, which can create dubious dependencies, are useful but, in practice, hard to find. So, we use this representation as an independence map, as an efficient and trustable model, which can be used to read off independencies, but we do not assume a lack of confounders.

Consequently, by identifying independencies in our Bayesian network, we can exclude diseases associated with but not strongly relevant to MDD. This helps us focus on genetic factors that are genuinely related to MDD, avoiding those pleiotropic effects that are not directly linked to the disorder.

We discussed further these limitations in the revised version of the main text (see Discussion, lines 507-509):

“Finally, our Bayesian network methodology is sensitive to unknown confounders and selection bias, potentially causing spurious correlations, but the fairly complete nature of our cohorts and the cross-cohort design mitigate this danger.”

- Line 671 p 31 – if “presence of one disease (e.g., MDD) directly influences the probability distribution of another” -> in my understanding, this is very close to being causal, although the authors say this is not. Could you further clarify this concept?

In the context of our Bayesian network, the directed arrows do indeed indicate that the presence of one disease can influence the probability distribution of another; however, this is fundamentally different from asserting a causal relationship for several reasons. We have clarified these aspects further in the revised version of the manuscript (see section “Assessing diseases strongly relevant to MDD” in the Online methods, Lines 703-714):

“We also note that while the relationships in our Bayesian network are direct and unmediated by other diseases, they do not necessarily imply causation. This directness refers to the absence of intermediate variables within the network’s model structure, distinguishing these relationships from mere correlations at the abstraction level defined by the entire set of variables in the analysis. However, direct probabilistic relationships in the Bayesian network are derived from observational data, not from interventional studies that manipulate one variable to directly observe its effect on another. Without the ability to control or manipulate the conditions, the relationships might still be influenced by unobserved confounding factors []. The direct relationships in the network are based on the strongest statistical dependencies observed in the data, but these dependencies alone do not fulfil all criteria required to establish causality, such as eliminating all potential confounders and demonstrating that the relationship is not reversible.”*

[Additional explanation not included in the main text] A directed arrow in a Bayesian network may simply capture an apparent relationship due to a third, unobserved variable influencing both diseases (e.g., certain genetic variants or a non-genetic risk factor). In causal analysis, such confounders need to be explicitly controlled or adjusted for, which is not inherently done in the construction of typical Bayesian networks.*

- The authors mentioned privacy-preserving Bayesian federated clustering as a methodology novelty. Could they further describe it in methods and highlight how it can be achieved?

Thank you for your inquiry about the privacy-preserving aspects of our Bayesian federated clustering methodology.

In the first step, each participating site independently computes the relevance scores (i.e., the posterior probabilities that various variables are strongly relevant with respect to MDD). Because these computations are based on aggregated data and do not involve transmitting or sharing individual disease onset information, the privacy of individual-level data is maintained. After the

local computations, these cohort-specific relevance scores are then averaged to derive cross-cohort relevance scores.

For each participant, we calculate a multimorbidity score for specific time intervals (see equation at top of page 34) by summing the cross-cohort relevance scores for the diseases they have experienced within those intervals. This results in an aggregated score that reflects the burden of multimorbidity related to MDD but does not reveal any specific disease details beyond what is aggregated across the cohort.

Finally, the aggregated multimorbidity scores from all participants are compiled into counts of occurrences at each site, which are then used as inputs for the clustering algorithm. This further aggregation ensures that the clustering is performed on non-identifiable data, thus preserving the privacy of individual participants.

This approach also lays the groundwork for a meta-analysis in which additional partners could share the same type of aggregated data publicly, akin to the sharing of GWAS summary statistics. This shared data could then be utilized in further research, enhancing our understanding and analysis capabilities across broader datasets.

We have summarized these aspects in the revised version of the manuscript (see section “Clustering of participants” in the Online methods, Lines 804-813).

- The authors presented or highlighted some sig genes in table 2, and possibly some other loci are not listed there. Could they discuss whether the differences in the sig. genes link up to the different disease profiles in the 7 clusters? Eg some genes are specifically related to certain diseases in a cluster.

Thank you for this comment. Our clarified, fundamental hypothesis states that the multimorbidities strongly relevant to MDD partially share its molecular background, suggesting that these disorders enrich genes that exhibit pleiotropy with respect to MDD. Importantly, the clusters are designed to reflect varying temporal trajectories of MDD-related multimorbidity burden over the lifespan rather than on the presence or prevalence of individual comorbid diseases. Each cluster represents an aggregated profile of multimorbidity, focusing on the dynamic course of MDD-related diseases. The differences in disease prevalences observed among clusters stem directly from these distinct temporal trajectories. Consequently, each cluster inherently enriches specific pleiotropic genes that share a common molecular basis with MDD and influence the particular pattern of multimorbidity trajectory that defines the cluster. In short, this approach is designed to identify those genetic variants that are not merely associated with individual comorbid conditions but are integral to the broader biological context of MDD-related multimorbidity trajectories. In other words, our work investigates the fundamental open question of the role of pleiotropy in MDD, for example, related to somatic diseases and a wide range of brain-body interactions.

We have demonstrated these genetic correlations between diseases and MDD-related clusters in Supplementary Fig. S13, and they are further discussed in the Results section (lines 318-322). This observation aligns with findings from network medicine and genetic association studies that highlight common biological mechanisms among comorbid disorders (Barabási et al. 2011, Menche et al 2015, Wang et al 2017, Brainstorm Consortium 2018). We have now illustrated this mechanism in Fig. 1, Rationale panel.

References:

Barabási AL et al. Network medicine: a network-based approach to human disease. *Nat Rev Genet.* 2011 Jan;12(1):56-68. PMID: 21164525; DOI: 10.1038/nrg2918

Menche J et al. Uncovering disease-disease relationships through the incomplete interactome. *Science* 347, 1257601 (2015). DOI: 10.1126/science.1257601

Wang K et al. Classification of common human diseases derived from shared genetic and environmental determinants. *Nat Genet.* 2017 Sep;49(9):1319-1325. PMID: 28783162; DOI: 10.1038/ng.3931

Brainstorm Consortium. Analysis of shared heritability in common disorders of the brain. *Science.* 2018 Jun 22;360(6395):eaap8757. PMID: 29930110; DOI: 10.1126/science.aap8757

- About software availability: May I know if the BN-BMLA software (for BN building) is publicly available as a package? If so, please list the link as well.

The BN-BMLA software used for building Bayesian Networks is currently available as standalone software written in C++, which is available at the github repository of the paper: <https://github.com/gezsi/mdd-clustering>. At present, no Python wrapper package is available. However, we are actively working on developing this feature and plan to release it along with an application note in the near future.

REVIEWERS' COMMENTS

Reviewer #2 (Remarks to the Author):

I congratulate the authors with a highly interesting and very impressive study; and I will particularly like to acknowledge the careful replies to the my concerns.

Reviewer #4 (Remarks to the Author):

I am satisfied with the authors' responses.

Reviewer #5 (Remarks to the Author):

My comments are mostly adequately addressed.

For the 1st comment on whether the network detects interactions, as pointed by the authors, "our current methodology centers on identifying and interpreting these structural relationships; meanwhile, it does not report quantitative interaction effects in the manner that traditional statistical models might." Pls kindly include an explanation of this limitation in the text and/or supp info, if not done yet.

Unique genetic and risk-factor profiles in clusters of major depressive disorder-related multimorbidity trajectories

NCOMMS-23-36313B

Response to Reviewers

Reviewer #5 (Remarks to the Author):

My comments are mostly adequately addressed.

For the 1st comment on whether the network detects interactions, as pointed by the authors, "our current methodology centers on identifying and interpreting these structural relationships; meanwhile, it does not report quantitative interaction effects in the manner that traditional statistical models might." Pls kindly include an explanation of this limitation in the text and/or supp info, if not done yet.

Dear Reviewer,

We sincerely appreciate your time, effort, and thoughtful consideration in reviewing our revisions. We have further clarified the aspect concerning interactions in the revised version of the manuscript (see section "Assessing diseases strongly relevant to MDD" in the Methods):

"We also note that in our Bayesian network framework, we focus on capturing structural probabilistic relationships between variables rather than quantifying interaction terms that occur in regression models. Although these interactions are quantified at the parametric level in Bayesian networks—through the conditional probability distributions of variables given others—our analyses primarily aimed to elucidate the structural relationships by performing exact Bayesian averaging over the parametric level rather than quantifying these interaction effects directly."